

# A robust chaos-inspired artificial intelligence model for dealing with nonlinear dynamics in wind speed forecasting

Caner Barış[1], Cağfer Yanarateş[2] and Aytaç Altan[1]

[1] Department of Electrical and Electronics Engineering, Zonguldak Bülent Ecevit University, Zonguldak, Turkey
[2] Department of Electrical and Energy, Kelkit Aydın Doğan Vocational School, Gümüşhane University, Gümüşhane, Turkey

## ABSTRACT

The global impacts of climate change have become increasingly pronounced in recent years due to the rise in greenhouse gas emissions from fossil fuels. This trend threatens water resources, ecological balance, and could lead to desertification and drought. To address these challenges, reducing fossil fuel consumption and embracing renewable energy sources is crucial. Among these, wind energy stands out as a clean and renewable source garnering more attention each day. However, the variable and unpredictable nature of wind speed presents a challenge to integrating wind energy into the electricity grid. Accurate wind speed forecasting is essential to overcome these obstacles and optimize wind energy usage. This study focuses on developing a robust wind speed forecasting model capable of handling non-linear dynamics to minimize losses and improve wind energy efficiency. Wind speed data from the Bandırma meteorological station in the Marmara region of Turkey, known for its wind energy potential, was decomposed into intrinsic mode functions (IMFs) using robust empirical mode decomposition (REMD). The extracted IMFs were then fed into a long short-term memory (LSTM) architecture whose parameters were estimated using the African vultures optimization (AVO) algorithm based on tent chaotic mapping. This approach aimed to build a highly accurate wind speed forecasting model. The performance of the proposed optimization algorithm in improving the model parameters was compared with that of the chaotic particle swarm optimization (CPSO) algorithm. Finally, the study highlights the potential of utilizing advanced optimization techniques and deep learning models to improve wind speed forecasting, ultimately contributing to more efficient and sustainable wind energy generation. This robust hybrid model represents a significant step forward in wind energy research and its practical applications.

## INTRODUCTION

The increase in the greenhouse effect caused by the consumption of fossil fuels is the basis of climate change, the effects of which are being felt more intensely around the world every

Corresponding author
Aytaç Altan, aytacaltan@beun.edu.tr

day. In order to effectively combat climate change, it has become imperative to reduce fossil fuel consumption and meet the required energy demand with clean energy sources (*Zhang et al., 2019b*). For this reason, the interest in clean energy sources, namely renewable energy sources, is increasing day by day around the world. Today, renewable energy sources, which account for about 10% of total global energy production, are expected to account for about 19% of global energy production by 2030 (*Kosana, Teeparthi & Madasthu, 2022*). Thanks to wind turbines, whose efficiency is increasing as technology advances, the share of wind energy among renewable energy sources is steadily increasing (*Wang et al., 2021*). According to 2022 data, approximately 40% of total renewable energy production is derived from wind energy. To effectively utilize wind energy, whose global installed capacity increased to 825 GW in 2022, it is essential to predict the wind speed, which has an uncertain and stochastic structure, with high accuracy (*BP, 2022*; *U.S. Energy Information Administration, 2022*; *Altan, Karasu & Zio, 2021*).

The non-stationary and highly variable nature of wind speed poses a significant challenge to the integration of wind energy into the electricity grid (*Chen et al., 2019*). This unpredictability adversely impacts the effective utilization of wind energy. To optimize the use of wind energy, it is crucial to identify regions with high wind energy potential and to achieve predictability of wind dynamics and speed. Therefore, it is necessary to create wind speed forecasting model structures that can cope with the nonlinear dynamics in the wind regime with chaotic and stochastic structure. The performance of the model structures depends on how robust and highly accurate the wind speed is estimated (*Wang et al., 2021*; *Altan, Karasu & Zio, 2021*). Given the nonlinear dynamics in wind speed time series, certain model structures that excel in long-term wind speed forecasting may struggle to achieve the same accuracy in short-term forecasts. Conversely, models that are effective in short-term wind speed forecasting may not perform as well in long-term forecasts (*Cai et al., 2019*).

Physical models forecast long-term wind speed based on geographical and meteorological parameters such as terrain, obstacles, topography, atmospheric pressure, and ambient temperature. While these models perform well in forecasting long-term wind speed, they struggle with short-term forecasting due to information lags (*Liu et al., 2012*). Although statistical models that integrate linear and nonlinear structures often outperform physical models in short-term wind speed forecasting, they assume that time series data are linear and stationary, which limits their effectiveness for non-stationary and nonlinear wind speed time series (*Liu, Ding & Bai, 2021*). Most existing wind speed forecasting methods use local, on-site data such as wind energy measurements and weather forecasts to optimize each region separately while also considering spatial-temporal interdependencies within wind energy production areas. The production of wind energy in each region exhibits autocorrelation. Energy production across different non-isolated wind farms in a region shows spatial cross-correlation with time lags. Therefore, a better understanding of the dynamic spatial-temporal interdependencies between wind farms in a region comprising tens or even hundreds of wind farms is crucial not only for achieving high accuracy wind speed forecasts, but also for seamless integration of wind energy into the electricity grid (*Zhao et al., 2018*).

The intrinsic dynamics of wind speed, including strong randomness, non-stationarity, nonlinearity, uncertainty, and volatility, pose significant challenges to short-term wind speed forecasting with high performance. To address these challenges, artificial intelligence-based models have been developed in recent years. These models include multi-layer perceptron (MLP), radial basis function (RBF), long short-term memory (LSTM), artificial neural network (ANN), recurrent neural network (RNN), back propagation neural network (BPNN), Elman neural network (ENN), wavelet neural network (WNN), convolutional neural network (CNN), and fuzzy logic (FL) models (*Ak et al., 2018*; *Zhang et al., 2016*; *Liu, Mi & Li, 2018*; *Ramasamy, Chandel & Yadav, 2015*; *Qian-Li et al., 2008*; *Wang et al., 2016*; *Liu et al., 2015*; *Xiao et al., 2017*; *Mehrkanoon, 2019*; *Damousis et al., 2004*; *Yu et al., 2018*; *Altan, Karasu & Zio, 2021*). To accurately forecast wind speed by capturing the non-linear characteristics of wind speed time series, these single model structures based on machine learning, ANNs and deep learning can be used, as well as hybrid model structures that use signal decomposition, feature extraction-selection and optimization techniques are used together with artificial intelligence algorithms are widely used (*Suo et al., 2023*; *Altan, Karasu & Zio, 2021*). The signal decomposition methods used in hybrid model structures designed for wind speed prediction decompose the wind speed time series into sub-bands and extract features that improve the prediction performance. Thanks to the extracted features, it is aimed to minimize the effect of nonlinear dynamics in wind speed time series within the model (*Liu et al., 2019*).

## Related works

*Lu et al. (2022)* conducted an analysis using the Pearson correlation coefficient (PCC) to determine the relationship between numerical weather prediction (NWP) variables and wind power. They selected NWP features with a PCC value greater than 0.5 to predict wind power. In another study, *Wu et al. (2022a)* used multiple correlation techniques, including PCC, partial correlation coefficient and maximum information coefficient (MIC), to select relevant weather factors. PCC and partial correlation coefficient were used to determine linear relationships between features and wind speed, while MIC was used to assess non-linear relationships. Unlike filter methods that pre-select features based solely on statistical measures, wrapper techniques evaluate the performance of different subsets of features directly on a prediction model and retain those that provide the best prediction accuracy. Recursive feature elimination (RFE), as discussed by *Houndekindo & Ouarda (2023)*, is a widely recognized wrapper method that iteratively trains the model and prunes features that do not contribute to improving prediction accuracy. In addition, evolutionary algorithms (EAs) are often used in conjunction with wrapper methods to efficiently search for optimal subsets of features. For example, *Liu et al. (2019)* applied a genetic algorithm to select input features for a prediction model based on the group method of data processing networks. Similarly, *Zhang et al. (2017)* implemented a backtracking search algorithm to identify relevant historical lags from decomposed sub-series and selected the features that provided the best accuracy for an extreme learning machine (ELM) model. In addition to the popular filter and wrapper methods, embedded feature selection approaches have also

gained traction. These methods involve training the prediction model on the full set of features and then ranking them based on their importance scores, with higher scoring features being prioritized. Random forest (RF) (*Niu et al., 2020*) and extreme gradient boosting (XGBoost) (*Zha et al., 2022*) are common models used for embedded feature selection.

*Liu et al. (2012)* developed a hybrid model consisting of empirical mode decomposition (EMD) and ANN algorithms to predict the next step in wind speed, using real wind speed time series from two different regions with intermittent sampling. The performance of the model was compared with that of single ANN and autoregressive integrated moving average (ARIMA) models. The hybrid model demonstrated its ability to effectively handle intermittent sampling in non-stationary wind series. *Ma et al. (2020)* proposed a novel hybrid model based on error correction, dual decomposition, and deep learning to improve wind speed forecasting accuracy. In the proposed complete ensemble empirical mode decomposition with adaptive noise (CEEMDAN)-error-variational mode decomposition (VMD)-LSTM model, CEEMDAN and VMD were applied to decompose the original wind speed series and error series, respectively, while LSTM was adopted to forecast each sub-series. The model was validated against four wind speed conditions and compared to other high-performing hybrid models such as ensemble empirical mode decomposition (EEMD)-error-LSTM and EEMD-error-VMD-LSTM. Among the models discussed in the study, the proposed CEEMDAN-error-VMD-LSTM model was reported to have the best forecasting performance. The model emphasizes that VMD effectively reduces the randomness and complexity of error series, allowing LSTM to forecast error series with higher accuracy. *Mi, Liu & Li (2019)* designed a novel multi-step hybrid wind speed forecasting model based on singular spectrum analysis (SSA), EMD, and CNN support vector machine (CNNSVM). SSA was employed to reduce the noise and extract the trend information from the original wind speed data. EMD was used to extract the fluctuation characteristics of the wind speed data, decomposing the wind speed time series into a series of sub-layers. Each wind speed sub-layer was then forecasted using the CNNSVM algorithm. To evaluate the forecasting performance of the proposed model, several models such as SVM, CNNSVM, EMD-BPNN, EMD-RBF, and EMD-ENN were used for comparison. The model was found to exhibit good forecasting accuracy and generalization performance in short-term multi-step wind speed forecasting, though it requires high-quality historical wind speed data. *Zhang et al. (2019a)* proposed a new model for short-term wind speed forecasting that combines the hybrid mode decomposition (HMD) technique and the online sequential outlier robust extreme learning machine. In the data pre-processing stage, the wind speed time series was deeply decomposed using HMD, which integrates VMD, sample entropy, and wavelet packet decomposition. The study highlights that HMD can significantly improve forecasting performance by reducing non-stationary characteristics in wind speed. The high complexity and strong non-linearity of wind speed data make capturing the characteristic features of the original time series challenging. *Liu, Ding & Bai (2021)* proposed a new wind speed forecasting model combining EMD with RNN and ARIMA algorithms to tackle the strong randomness and non-linear dynamics in wind speed. The proposed

EMD-LSTM-ARIMA model uses EMD to decompose the original wind speed series, LSTM to predict complex high frequency sub-series, and ARIMA to predict stable low frequency sub-series. The model was validated with four wind speed data samples and compared with the results of single LSTM, single gated recurrent unit (GRU), EMD-LSTM, EMD-GRU, and EMD-GRU-ARIMA models. The study found that hybrid models outperformed single models in terms of forecast performance and that decomposition methods effectively improve wind speed forecasting. However, it also reported that decomposition methods are not suitable for all wind speed data sets and models. Using a single model for forecasting after decomposition can increase errors in low-frequency sub-series forecasts. The study addressed this issue by using ARIMA to forecast low frequency sub-series and residuals and LSTM to forecast high frequency sub-series with high entropy.

In the optimization of ensemble models, multi-objective optimization (MOO) algorithms play a crucial role in determining the optimal weight distribution among the component models. *Zhou, Wang & Zhang (2020)* proposed an ensemble prediction approach that integrates five different models, whose combination weights are simultaneously optimized using multi-objective particle swarm optimization (MOPSO). *Sarangi, Dash & Bisoi (2023)* introduced a new variant of the particle swarm optimization algorithm that integrates a sine-cosine mechanism to optimize short-term wind speed forecasts using a deep belief network and a multi-core random vector functional link network. *Wang et al. (2022)* similarly employed a multi-objective grey wolf optimization (MOGWO) algorithm to optimize the weight distribution across multiple neural networks in an ensemble setup. While MOO algorithms have been less frequently applied to feature selection in wind energy forecasting, some studies, such as those by *Lv & Wang (2022, 2023)*, used the non-dominated sorting genetic algorithm II (NSGA-II) to fine-tune meteorological feature selection by simultaneously reducing the number of features and minimizing forecast errors.

In hybrid modelling, the original wind speed data is typically divided into sub-series with different frequency characteristics. Each subseries is then modelled individually and the resulting predictions are combined to form the final forecast (*Sun, Zhao & Zhao, 2022*). For example, *Li et al. (2022)* used VMD to decompose wind speed data into intrinsic mode functions (IMFs) at different frequencies, with each IMF predicted using a bidirectional LSTM model. *Wu et al. (2022b)* used EEMD to extract IMFs from wind speed data, and these multi-dimensional sub-series were modelled directly using a transformer network. Hybrid models that combine signal decomposition algorithms with machine learning and deep learning techniques often perform better in wind speed forecasting than single models. However, their ability to capture non-linear dynamics in wind speed time series is still limited. To overcome this problem, optimization algorithms are frequently used in hybrid wind speed forecasting models to improve model parameters and select characteristic features. *Chen et al. (2018b)* proposed a new method called EnsemLSTM for wind speed forecasting using an ensemble of LSTMs, support vector regression machine (SVRM), and an extremal optimization (EO) algorithm. The approach aimed to avoid the weak generalization and robustness drawbacks of a single deep learning approach when

dealing with diverse data by using a set of LSTMs with different hidden layers and neurons to uncover and utilize latent information in wind speed time series. Predictions from the LSTMs were combined in a nonlinear learning regression upper layer composed of SVRM, with the parameters of the upper layer optimized by the EO algorithm. Experimental results showed that the proposed model outperformed other forecasting models without optimization algorithms in terms of prediction performance. *Lv & Wang (2022)* proposed an effective combined model system for wind speed forecasting. They developed a hybrid time series decomposition (HTSD) strategy to simultaneously extract linear patterns and frequency domain features from raw wind speed data. The decomposition parameters were optimized using the multi-objective binary backtracking search algorithm (MOBBSA). An advanced sequence-to-sequence (Seq2Seq) forecaster was used to handle component series uniformly, with the final results averaged from several different Seq2Seq model predictions. The proposed model achieved an average percentage improvement of 59.92% over the latest techniques. *Suo et al. (2023)* developed a hybrid wind speed forecasting model using time varying filtering-based EMD (TVFEMD), fuzzy entropy (FE), partial autocorrelation function (PACF), improved chimp optimization algorithm (IChOA), and bi-directional gated recurrent unit (BiGRU). TVFEMD decomposed the original wind speed data into mode components, and FE aggregation was used to reduce computational complexity. PACF processed the components to extract key input features. BiGRU parameters were optimized using IChOA, an advanced version of the chimp optimization algorithm, which improved forecasting accuracy. The forecasted components were combined to produce the final prediction. The study highlighted the effectiveness of IChOA in optimizing BiGRU parameters and its role in improving model performance. *Altan, Karasu & Zio (2021)* proposed a hybrid short-term wind speed forecasting model that combines signal decomposition, deep learning, and optimization algorithms. They used the improved complete ensemble empirical mode decomposition with adaptive noise (ICEEMDAN) technique to obtain each IMF and created a forecasting model using LSTM neural networks for each IMF. The weights of each output were optimized using the grey wolf optimizer (GWO) algorithm. The optimization algorithm improved the model's performance by approximately 20%.

RNNs are widely used in wind speed forecasting due to their ability to capture temporal dependencies in sequential data. The most commonly used RNN models are LSTM and GRU. LSTM networks use a unique cell structure with gating mechanisms, making them well suited to preserving long-term dependencies while mitigating problems such as gradient vanishing. GRU, a simplified variant of LSTM, also effectively models long-and short-term dependencies with a more streamlined gating structure. In their study, *Sun, Zhao & Zhao (2022)* improved short-term wind speed predictions by integrating LSTM, GRU and a deep belief network. Recent research has explored more advanced RNN architectures, including bidirectional networks (*Li et al., 2022*), Seq2Seq models (*Lv & Wang, 2022*), and attention mechanisms (*Tian, Niu & Wei, 2022*). In contrast to RNNs, CNNs excel at identifying spatial features within input data. Given the complementary strengths of RNNs and CNNs, recent studies have investigated hybrid approaches to improve predictive performance. Two main combination strategies have emerged. The

first involves the sequential stacking of CNN and RNN layers to form a cascade network (CNN-RNN). For example, *Yu et al. (2023)* integrated a CNN with an enhanced LSTM to predict wind speed, where the CNN extracted temporal correlations and the modified LSTM captured time-frequency dynamics. The second approach introduces convolutional operations directly into the LSTM architecture, resulting in the convolutional LSTM (ConvLSTM) model developed by *Neshat et al. (2022)*. This model is tailored for the processing of three-dimensional sequential data and allows the extraction of local correlations between different temporal features. The ConvLSTM model has been widely adopted in wind energy forecasting, demonstrating superior performance compared to many standard deep learning models (*Neshat et al., 2022*; *Liu, Ding & Bai, 2021*; *Chen et al., 2018b*).

Table 1 provides an overview of the literature discussed, offering a concise analysis of the main methods, descriptions, and features in wind speed forecasting studies. This includes research on feature selection techniques, MOO algorithms, and hybrid deep learning models.

Although the proposed approaches have reduced the error in wind speed forecasting to some extent, the nonlinear and chaotic nature of wind speed series, together with their instability and high noise, make it difficult to achieve high prediction accuracy. Hybrid models that combine the strengths of individual models are being developed to address this problem, but parameter uncertainty in these models can make them impractical for engineering applications. Therefore, robust forecasting models that effectively deal with the nonlinear and chaotic characteristics of wind speed series are needed. In this study, a new wind speed forecasting model is developed using robust EMD (REMD) and LSTM algorithms optimized with the African vultures optimization (AVO) algorithm based on tent chaotic mapping. The model aims to address the non-linear characteristics of wind speed, such as non-stationarity, strong randomness, chaos, uncertainty, and volatility, to achieve high short-term forecasting accuracy. By using REMD, we improve forecast reliability and provide a robust solution to the inherent complexity of wind speed time series. This hybrid approach combines LSTM's sequential learning capabilities with AVO's robust optimization techniques to effectively deal with the complexities of wind speed variability and randomness. The model aims to provide more accurate and stable wind speed forecasts, which are critical for optimizing wind energy production and supporting meteorological applications. The remainder of this study is structured as follows. The second section introduces the components of the developed hybrid wind speed forecasting model, including the robust signal decomposition technique, the LSTM neural network, and the improved African vulture optimization algorithm based on tent chaotic mapping and time-varying mechanism (TAVO). The third section presents and discusses the experimental results of the forecasting model using an hourly wind speed dataset collected from the Bandırma meteorological station, a location known for its high wind energy potential in Turkey. The results and future projections of the study are highlighted in the conclusion section.

**Table 1 Overview of the literature review.**

| Modules | Categories | Typical methods | Descriptions | Characteristics |
|---|---|---|---|---|
| Feature selection approaches | Filter | PCC (*Lu et al., 2022*; *Wu et al., 2022a*) | Analyze the relationships among the variables, and select the features that have strong correlations with the dependent variable. | Simple to implement; quick execution time; disregards the impact of features on the prediction model. |
| | | PACF (*Suo et al., 2023*) | | |
| | | MIC (*Wu et al., 2022a*) | | |
| | Wrapper | RFE (*Houndekindo & Ouarda, 2023*) | Choose the subset of features according to their accuracy in the prediction model. | Take into account the impact of features on the model; capable of easily generating a high-accuracy feature subset |
| | | EA (*Liu et al., 2019*; *Zhang et al., 2017*) | | |
| | Embedded | RF (*Niu et al., 2020*) | Build a model that can assign importance scores to the evaluated features, with the top-scoring features being chosen. | Account for the impact of features on the model without requiring iterative training. |
| | | XGBoost (*Zha et al., 2022*) | | |
| MOO algorithms | Parameter tuning; feature selection | GWO (*Altan, Karasu & Zio, 2021*) | Optimize multiple objective functions at the same time using evolutionary techniques, and evaluate the solutions based on Pareto optimality. | Various optimization objectives can be addressed, resulting in a set of solutions where none are worse than the others. |
| | | MOGWO (*Wang et al., 2022*) | | |
| | | MOPSO (*Zhou, Wang & Zhang, 2020*) | | |
| | | NSGA-II (*Lv & Wang, 2022*; *Lv & Wang, 2023*) | | |
| Hybrid deep learning models | Decomposition methods | EMD (*Liu et al., 2012*) | Decompose the original time series into several simpler subseries with varying frequencies. | Facilitates the extraction of intrinsic local features from raw data and the reduction of its noise. |
| | | EEMD (*Wu et al., 2022b*) | | |
| | | VMD (*Sun, Zhao & Zhao, 2022*; *Li et al., 2022*) | | |
| | | CEEMDAN (*Ma et al., 2020*) | | |
| | | HMD (*Zhang et al., 2019a*) | | |
| | | ICEEMDAN (*Altan, Karasu & Zio, 2021*) | | |
| | Forecasting models | LSTM/GRU (*Sun, Zhao & Zhao, 2022*) | RNNs with gate-based control mechanisms. | Capable of learning both short-term and long-term dependencies in sequences. |
| | | Attention (*Tian, Niu & Wei, 2022*) | Apply varying weights to the input features. | Identifies and emphasizes critical influencing factors; provides interpretability. |
| | | Seq2Seq (*Lv & Wang, 2022*) | A combined deep learning architecture featuring encoding and decoding networks. | Adaptable to sequences of varying lengths. |
| | | CNN-RNN (*Yu et al., 2023*; *Mi, Liu & Li, 2019*) | A cascading configuration of CNN and RNN. | Extracts local and temporal features concurrently. |
| | | ConvLSTM (*Neshat et al., 2022*; *Liu, Ding & Bai, 2021*; *Chen et al., 2018b*) | Incorporate convolution operations within the LSTM cell. | |

# PROPOSED METHODOLOGY

This section outlines the methods used to develop the REMD-LSTM-TAVO model for wind speed forecasting. The model integrates data pre-processing, the REMD technique, the LSTM neural network, and the TAVO algorithm. Specifically, the model uses an LSTM

neural network for each IMF obtained by REMD and optimizes the weight coefficients of each output with TAVO. This comprehensive approach aims to address the complex and non-linear nature of wind speed data to provide a robust and accurate forecast model.

## Robust empirical mode decomposition

The EMD method was developed by *Huang et al. (1998)* to process linear and non-stationary signals. Its primary purpose is to decompose a non-stationary signal using the Hilbert-Huang transformation until the signal becomes stationary. This method decomposes complex signals into relatively stable IMFs based on the assumption that signals have many oscillatory modes simultaneously. These IMFs are extracted from the original time series based on local characteristic scales. To qualify as an IMF, a function must satisfy two criteria: the number of extrema and zero crossings should be approximately equal, with a maximum difference of one, and the mean of the envelope defined by the local maxima and minima must be zero (*Chen, Lai & Yeh, 2012*). This ensures that the IMF is nearly periodic and its mean is adjusted to zero.

Given an original wind speed time series $z(t)$, the EMD algorithm can be described mathematically as:

$$z(t) = \sum_{j=1}^{m} f_j(t) + R_m(t) \tag{1}$$

Here, $f_j(t)$ $(j = 1, 2, \ldots, m)$ represents different IMFs, while $R_m(t)$ is the residual after deriving the $m$-th IMF. The process of extraction of each IMF includes:

i) Identifying all extrema in the original wind speed time series $z(t)$.
ii) Calculating the upper $\{z_u(t)\}$ and lower $\{z_l(t)\}$ envelopes using cubic spline interpolation.
iii) Calculating the mean envelope $m(t) = \frac{(z_u(t) + z_l(t))}{2}$.
iv) Subtracting the mean envelope from the original wind speed time series to obtain $d(t) = z(t) - m(t)$.
v) Testing the characteristics of $d(t)$. If $d(t)$ qualifies as an IMF, $f(t)$ is set to $d(t)$ and designated as the $i$-th IMF. The residual $z(t)$ is updated as $R(t) = z(t) - f(t)$. If $d(t)$ does not meet the criteria for an IMF, $z(t)$ is replaced with $d(t)$.
vi) (i) to (v) are repeated until the stop conditions are met.

REMD is an enhanced version of EMD that aims to handle non-linear and non-stationary signals with increased robustness to noise and outliers (*Chen et al., 2018a*). The REMD technique introduces bidirectional filtering to suppress noise, and its algorithm consists of the following steps and mathematical expressions:

**Step 1:** Identify local maxima and minima: In the first step, identify the local maxima and minima points in the time series.

**Step 2:** Smooth the estimated envelope: Use the Eq. (2) to calculate the weights for smoothing the estimated envelope:

$$\omega_{\{m,n\}}[k,j] \propto \exp\left(-\frac{\Delta I[m,\ n;\ k,\ j]^2}{2r\sigma_r^2} - \frac{m^2 - n^2}{2r\sigma_s^2}\right). \tag{2}$$

Here, $\Delta I[m,\ n;\ k,\ j]$ represents the intensity difference between pixel pairs.

**Step 3:** Create maximum and minimum envelopes:

Maximum envelope $E_{max}$ and minimum envelope $E_{min}$ are created by solving the Eq. (3):

$$E^* = \left[P_e^T P_e + \lambda(D - W)^T(D - W)\right]^{-1} P_e^T P_e I. \tag{3}$$

**Step 4:** Calculate the mean envelope: Calculate the mean envelope as

$$\overline{E_{(i)}} = \frac{E_{max} + E_{min}}{2}. \tag{4}$$

**Step 5:** Extract IMF function: Calculate the IMF function using

$$h_i^c = I_{(i)} - \overline{E_{(i)}}. \tag{5}$$

**Step 6:** Repeat steps to meet IMF requirements: Repeat the previous steps (ii) through (v) until the requirements for an IMF are met.

**Step 7:** Determine modulation mixing issues: Identify any modulation mixing issues using the Eq. (5).

**Step 8:** Create adaptive mask signal: Use the Eq. (6) to create the adaptive mask signal $M_{COS}$

$$h_i = \frac{h_i^+ + h_i^-}{2}. \tag{6}$$

**Step 9:** Calculate residual: Calculate the residual as $R_{es} = I_{(i)} - h_i$. If there is no additional point, the EMD process stops; otherwise, update $i = i + 1$ and revert $I_{(i)} = R_{es}$. The final IMF components $\{h_i\}$ and the residual $R_{es}$ are given as output. After the above nine steps, the $Y$ sedimentation series is expressed as

$$Y = \sum_{i=1}^{m} h_i + R_{es}. \tag{7}$$

## Long short-term memory network

LSTM is an artificial recurrent neural network (RNN) architecture that has been widely used in deep learning. LSTM networks have been specifically designed to overcome the limitations of RNNs in modelling long-term dependencies. In essence, LSTMs are a specialized variant of RNNs. RNNs are neural networks capable of learning sequential patterns by processing input sequences through internal loops. Within an RNN structure, backpropagation is used to learn weights, and gradients are propagated using the chain

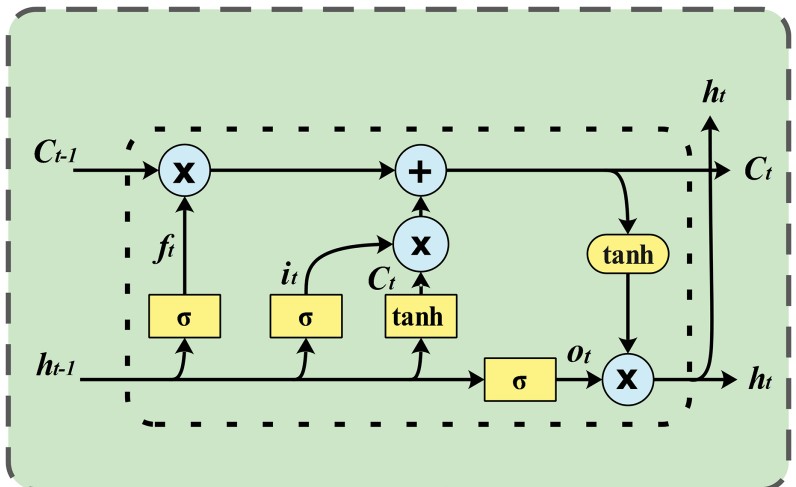

**Figure 1 LSTM network architecture.** LSTM models sequential data, such as wind speed data, using a type of recurrent neural network architecture. Its architecture includes a memory cell and gates that control the flow of information, allowing it to learn long-range dependencies.

rule. However, when gradients are backpropagated through activation functions such as sigmoid and tanh, they can become excessively small (or excessively large), leading to the vanishing (or exploding) gradient problem. This problem makes backpropagation vulnerable to long-range dependencies. LSTM models have been developed to mitigate these challenges, providing a stable and robust solution to both long-and short-term dependency problems. An LSTM network not only preserves adjacent temporal information, but also effectively manages long-term information (*Hochreiter & Schmidhuber, 1997*).

A standard LSTM unit consists of a cell, an input gate, an output gate, and a forget gate. The cell is responsible for remembering values over varying time intervals, while the three gates regulate the flow of information into and out of the cell. LSTM networks are well suited for classifying, processing and making predictions based on time series data because they can handle unknown time delays between important events in a time series. LSTMs were developed to address the exploding and vanishing gradient problems that can occur when training traditional RNNs. The architecture of an LSTM network is illustrated in Fig. 1.

LSTMs consist of memory blocks instead of neurons. In an LSTM network, each memory cell is accompanied by input, output, and forget gates. These gates enable the LSTM network to add or remove information from the cell state. The memory cell is considered the core of the LSTM network. New information that can be accumulated in the cell is defined by multiplying the input data by the output of the input gate. The propagated information within the network is calculated by multiplying it by the activation of the output gate. The cell state from the previous time step is multiplied by the activation of the forget gate to determine whether the cell's previous state should be forgotten. The

procedure for updating the cell state and computing the LSTM network output is carried out through the following steps (*He et al., 2019*):

i) Decide which information to discard from the cell state: the values of $x_t$ and $h_{t-1}$ are passed through a sigmoid function to determine what should be discarded.

$$f_t = \sigma\left(w_f \cdot [h_{t-1}, x_t] + b_f\right). \tag{8}$$

ii) Determine what new information will be stored in the cell state: Initially, a sigmoid layer determines the information to retain in the cell state. Subsequently, the values from $x_t$ and $h_{t-1}$ passed through the $tanh(\cdot)$ layer are regarded as a new candidate value $\tilde{C}_t$.

$$i_t = \sigma(w_i \cdot [h_{t-1}, x_t] + b_i) \tag{9}$$

$$\tilde{C}_t = tanh(w_C \cdot [h_{t-1}, x_t] + b_C). \tag{10}$$

iii) Update the previous cell state $C_{t-1}$ to the new cell state $C_t$: multiply the previous cell state $C_{t-1}$ by $f_t$ to forget the information we decided to forget, and plus $i_t$ times $\tilde{C}_t$ to get a new cell state $C_t$.

$$C_t = f_t * C_{t-1} + i_t * \tilde{C}_t. \tag{11}$$

iv) Determine the information to be output: Initially, a sigmoid layer determines which information will be output from the cell state. Then, apply the cell state $C_t$ to a $tanh(\cdot)$ function and multiply it by the output of the sigmoid gate.

$$o_t = \sigma(w_o \cdot [h_{t-1}, x_t] + b_0 \tag{12}$$

$$h_t = o_t * tanh(C_t). \tag{13}$$

Here $b_f$, $b_i$, $b_C$, and $b_0$ represent the bias vectors, while $w_f$, $w_i$, $w_C$, and $w_o$ represent the weight matrices. $\sigma(\cdot)$ represents the sigmoid function used as the gate activation function, while $tanh(\cdot)$ denotes the hyperbolic tangent function used as the input and output block activation functions, as shown below:

$$\sigma(x) = \frac{1}{1 + e^{-x}} \tag{14}$$

$$tanh(x) = \frac{e^x - e^{-x}}{e^x + e^{-x}}. \tag{15}$$

### Chaotic African vultures optimization

Based on different design inspirations, metaheuristic optimization algorithms are classified into single solution-based and population-based algorithms. In single solution-based metaheuristic optimization, only one solution participates in the optimization process, leading to an insufficient exploration of the entire solution space and making the algorithm prone to local optima. On the other hand, population-based metaheuristic algorithms

involve a group of solutions in the optimization process. This allows individuals within the population to explore a wider solution space and share information, reducing the likelihood of getting stuck in local optima. In particular, nature-inspired metaheuristic optimization algorithms are adept at balancing the exploration and exploitation phases of the optimization process. They leverage their exploration capabilities to avoid local optima and their exploitation capabilities to ensure that each solution converges to a better objective. As a result, these algorithms have been proposed and used extensively in recent years due to their effectiveness in optimizing complex problems (*Dominico & Parpinelli, 2021*; *Altan, Karasu & Bekiros, 2019*; *Kashani et al., 2022*; *Altan, 2020*; *Mirjalili & Lewis, 2016*; *Altan, Karasu & Zio, 2021*; *Yang, 2020*; *Yağ & Altan, 2022*).

Among these, the AVO algorithm stands out for its comprehensive exploration and exploitation mechanisms. The use of a random strategy enhances both the exploration capability of the exploitation mechanism and the exploitation capability of the exploration mechanism. This dual enhancement ensures that the AVO algorithm not only avoids local optima and achieves rapid convergence, but also maintains a balanced level of divergence. However, despite the AVO algorithm's efforts to balance exploration and exploitation capabilities in its design, there are still three main shortcomings. First, while the exploitation mechanism speeds up convergence in the early exploration phase, it can hinder the individual's ability to perform a comprehensive global search within the solution space. Without thorough global exploration, the AVO algorithm is likely to settle on a locally optimal solution in later stages. Second, during the exploration phase, AVO algorithm relies solely on the information from the top two individuals in the population, neglecting the individual's own information. This results in a slow convergence rate in the early stages. Therefore, for problems that require low time consumption or high real-time performance, the AVO algorithm may struggle to find an adequate solution. Third, in the later exploitation stage, the AVO algorithm assumes that the first best solution and second-best solution have the same impact on other individuals. This assumption does not effectively balance the AVO algorithm's exploration and exploitation capabilities, resulting in insufficient exploration in the early stages and insufficient exploitation in the later stages. To address the three aforementioned shortcomings of AVO, this study uses the TAVO algorithm. First, to enhance TAVO's global exploration capability in the early stages, tent chaos is used to initialize the population. This ensures that each individual is more evenly distributed in the solution space during initialization, thereby improving TAVO's exploration capability. Second, TAVO records each individual's locally optimal solution to update positions during the exploration phase. This approach leverages individual historical information to enhance TAVO's local exploitation ability, allowing for the rapid identification of good feasible solutions. Additionally, two time-varying coefficients are introduced that change with the number of iterations. One coefficient decreases with the number of iterations and measures the influence of the best individual on the current individual. The other coefficient increases with the number of iterations and measures the influence of the individual historical optimization on the current individual. This strategy balances the exploration and exploitation capabilities of the algorithm. Furthermore, in the exploitation phase of TAVO, two additional time-varying coefficients

are designed. Similar to the previous coefficients, one decreases with the number of iterations and measures the influence of the best individual on the current individual. The other increases with the number of iterations and measures the influence of the second-best individual on the current individual. This dual-coefficient approach ensures robust exploration in the early stages and effective exploitation in the later stages, enabling TAVO to achieve superior results (*Fan, Li & Wang, 2021*).

AVO algorithm has been developed by simulating and modelling the foraging behaviour and living habits of African vultures. In AVO, these behaviors and habits are simulated according to the following criteria:

i) The African vulture's population consists of $N$ vultures, with $N$ determined by the algorithm user based on the specific context. Each vulture operates in a $D$-dimensional position space, where $D$ corresponds to the dimensions of the problem being addressed. Additionally, a maximum number of iterations $T$ must be predefined, reflecting the maximum number of actions a vulture can perform. Thus, the position of each vulture $i$ (where $1 \leq i \leq N$) at different iterations $t$ (where $1 \leq t \leq T$) is represented by

$$X_i(t) = [x_{i1}(t), x_{i2}(t), \ldots, x_{iD}(t)]. \tag{16}$$

This equation represents the position vector of each vulture in the population at iteration $t$

ii) Based on the living habits of African vultures, the population is divided into three groups. If the fitness value of the feasible solution measures the quality of a vulture's position, the first group consists of the vultures with the best feasible solution. The second group contains those with the second-best feasible solution. The remaining vultures form the third group.

iii) Vultures forage collectively as a population, with different types of vultures playing distinct roles within the group.

iv) Assuming that the fitness value of the feasible solution indicates the relative quality of vultures, the weakest and hungriest vultures correspond to those with the poorest fitness values. Conversely, the strongest and most satiated vulture represents the one with the best fitness value. In AVO, all vultures strive to approach the best vultures and distance themselves from the worst.

Based on the aforementioned four principles, AVO simulates various vulture behaviors during the foraging process by dividing the problem-solving procedure into five distinct stages.

***Stage 1:*** *Population grouping*

As per the second rule, following initialization or prior to commencing the next action, vultures must be categorized based on their quality. The vulture representing the best solution is assigned to the first group, while the one representing the second-best solution is placed in the second group. The remaining vultures are assigned to the third group.

Given that both the top two vultures have a guiding influence, Eq. (17) is formulated to determine which vulture the others should move towards in the current iteration.

$$R_i^t = \begin{cases} BestVulture_1^t, & p_i^t = L_1 \\ BestVulture_2^t, & p_i^t = L_2 \end{cases} \tag{17}$$

where $BestVulture_1^t = \begin{bmatrix} b_{11}^t, & \ldots, & b_{1d}^t, \ldots, & b_{1D}^t \end{bmatrix}$ represents the best vulture, $BestVulture_2^t = \begin{bmatrix} b_{21}^t, & \ldots, & b_{2d}^t, \ldots, & b_{2D}^t \end{bmatrix}$ represents the second best vulture, $L_1$ and $L_2$ are two random numbers in the range $[0, 1]$ whose sum is 1. $p_i^t$ is determined using the roulette wheel strategy, and its calculation is given by

$$p_i^t = \frac{f_i^t}{\sum\limits_{i=1}^{m} f_i^t} \tag{18}$$

where $f_i^t$ stands for the fitness value of the vultures in the first and second groups, while $m$ denotes the total count of vultures in these groups.

**Stage 2:** *The hunger of vultures*

If a vulture isn't very hungry, it possesses enough strength to venture farther in search of food. Conversely, if a vulture feels particularly hungry, it lacks the physical stamina to sustain a long flight. Consequently, hungry vultures become more aggressive, preferring to remain near vultures with food rather than forage independently. Thus, vulture behavior can be categorized into exploration and exploitation stages based on hunger levels. The degree of hunger serves as an indicator for the transition from the exploration stage to the exploitation stage. The hunger degree $F_i^t$ of the $i^{th}$ vulture at the $t^{th}$ iteration can be calculated by

$$F_i^t = \left( 2 \times rand_{i1}^t + 1 \right) \times z^t \times \left( 1 - \frac{t}{T} \right) + g^t \tag{19}$$

where $rand_{i1}^t$ is a randomly generated number within the interval $[0, 1]$, $z^t$ is a random number within the range of $[-1, 1]$, and $g^t$ is computed by

$$g^t = h^t \times \left( sin^k \left( \frac{\pi}{2} \times \frac{t}{T} \right) + cos \left( \frac{\pi}{2} \times \frac{t}{T} \right) - 1 \right) \tag{20}$$

where $h^t$ is a randomly generated number within the range of $[-2, 2]$, and $k$ is a predefined parameter that indicates the likelihood of the vulture transitioning to the exploitation stage. A higher value of $k$ suggests a greater probability of transitioning to the exploration stage in the final optimization phase. Conversely, a lower value of $k$ indicates a higher likelihood of transitioning to the exploitation stage in the final optimization phase.

Based on the design principle of the equation, $F_i^t$, gradually decreases as the number of iterations increases, and the rate of decrease continues to escalate. Hence, when $\left| F_i^t \right|$ exceeds 1, vultures engage in the exploration stage, seeking new food across various regions. Conversely, when $\left| F_i^t \right|$ is less than 1, vultures transition to the exploitation stage, searching for superior food sources in the vicinity.

**Stage 3:** *Exploration stage*

In nature, vultures possess exceptional eyesight, enabling them to efficiently locate food and dying animals. Hence, when seeking sustenance, vultures initially assess their surrounding environment before embarking on extensive flights to locate food sources (*Sasmal et al., 2024*). In AVO, two exploration behaviors are implemented, with a parameter $p_1$ used to determine the specific behavior the vulture will exhibit. This parameter $p_1$ is set during the algorithm's initialization and ranges from 0 to 1. AVO selects the vulture's exploration method based on a random number within the range $[0, 1]$, comparing it to $p_1$. The vulture's exploration phase is represented by

$$X_i^{t+1} = \begin{cases} R_i^t - D_i^t \times F_i^t, & p_1 \geq rand_{p1}^t \\ R_i^t - F_i^t + rand_{i2}^t \times \left((ub - lb) \times rand_{i3}^t + lb\right), & p_1 < rand_{p1}^t \end{cases} \quad (21)$$

where $X_i^{t+1}$ denotes the position of the $i^{th}$ vulture at the $(t+1)^{th}$ iteration, $rand_{p1}^t$, $rand_{i2}^t$, and $rand_{i3}^t$ are random values uniformly distributed within the range $[0, 1]$, $R_i^t$ is determined by Eq. (17), $F^t$ is calculated using Eq. (19), $ub$ and $lb$ denote the upper and lower bounds of the solution space, and $D_i^t$ is computed using

$$D_i^t = \left| C \times R_i^t - X_i^t \right| \quad (22)$$

to represent the distance between the vulture and the current optimal position. Here, $X_i^t$ denotes the position of the $i^{th}$ vulture at iteration $t$, and $C$ is a random value uniformly distributed between 0 and 2.

**Stage 4:** *Exploitation stage (medium)*

To maintain a balance between exploration and exploitation and prevent a rapid transition in the algorithm's middle phase, vultures enter the medium-term exploitation stage when the value of $\left|F_i^t\right|$ is between 0.5 and 1. In this phase, a parameter $p_2$, ranging from 0 to 1, determines whether the vulture engages in food competition or rotating flight. A random number $rand_{p2}^t$ within the $[0, 1]$ range is generated before the vultures act. If $rand_{p2}^t$ is equal to or greater than $p_2$, the vultures will compete for food. If $rand_{p2}^t$ is less than $p_2$, they will perform a rotating flight maneuver.

*i. Food competition*

When $\left|F_i^t\right|$ falls between 0.5 and 1, this signifies that the vulture is sufficiently nourished and vigorous. In this state, stronger vultures tend to withhold food from others, while weaker ones congregate and attempt to challenge the stronger vultures for resources. The position update formula for vultures in this scenario can be given as

$$X_i^{t+1} = D_i^t \times \left(F_i^t + rand_{i4}^t\right) - d_i^t \quad (23)$$

where $rand_{i4}^t$ is a random variable uniformly distributed within $[0, 1]$, and $d_i^t$ is defined by

$$d_i^t = R_i^t - X_i^t \quad (24)$$

*ii. Rotating flight*

When a vulture is sufficiently nourished and vigorous, it exhibits competitive behavior for food and tends to hover at high altitudes. AVO employs a spiral model to simulate this hovering behavior. Consequently, the position update equation for vultures during this rotating flight can be described as

$$X_i^{t+1} = R_i^t - \left(S_{i1}^t + S_{i2}^t\right) \tag{25}$$

where $S_{i1}^t$ and $S_{i2}^t$ are defined as

$$S_{i1}^t = R_i^t \times \left(\frac{rand_{i5}^t \times X_i^t}{2\pi}\right) \times cos\left(X_i^t\right) \tag{26}$$

$$S_{i2}^t = R_i^t \times \left(\frac{rand_{i6}^t \times X_i^t}{2\pi}\right) \times sin\left(X_i^t\right). \tag{27}$$

Here, $rand_{i5}^t$ and $rand_{i6}^t$ are random values uniformly distributed between 0 and 1.

**Stage 5:** *Exploitation stage (later)*

When the value of $\left|F_i^t\right|$ falls below 0.5, most vultures in the population are satiated, but the two best-performing vultures become weak and hungry due to extended exertion. Consequently, vultures will engage in aggressive behaviors toward food, with many gathering at the same food source. In the latter exploitation phase, a parameter $p_3$ within the range $[0, 1]$ determines whether vultures perform attack or aggregation behaviors. Upon entering this phase, a random number $rand_{p3}^t$ within $[0, 1]$ is generated to decide their actions. If $rand_{p3}^t$ is equal to or greater than $p_3$, vultures display aggregation behavior. Conversely, if $rand_{p3}^t$ is less than $p_3$, they engage in attack behavior.

*i. Aggregation behavior*

In the final stages of AVO, as vultures have consumed a substantial amount of food, they gather around remaining food sources, leading to competition. The position update formula for vultures at this stage is given by

$$X_i^{t+1} = \frac{A_{i1}^t + A_{i2}^t}{2} \tag{28}$$

where $A_{i1}^t$ and $A_{i2}^t$ are determined by

$$A_{i1}^t = BestVulture_1^t - \frac{BestVulture_1^t \times X_i^t}{BestVulture_1^t - \left(X_i^t\right)^2} \times F_i^t \tag{29}$$

$$A_{i2}^t = BestVulture_2^t - \frac{BestVulture_2^t \times X_i^t}{BestVulture_2^t - \left(X_i^t\right)^2} \times F_i^t. \tag{30}$$

*ii. Attack behavior*

Similarly, in the late stage of AVO algorithm, vultures move towards the best vulture to access the remaining food. The position update formula in this phase is expressed as

$$X_i^{t+1} = R_i^t - \left|d_i^t\right| \times F_i^t \times Levy(dim) \tag{31}$$

Here, $dim$ represents the dimensionality of the solution and $Levy(\cdot)$ denotes the Lévy flight, calculated as

$$Levy(dim) = 0.01 \times \frac{r_1 \times \sigma}{|r_2|^{\frac{1}{\delta}}} \tag{32}$$

where $r_1$ and $r_2$ are random variables uniformly distributed within $[0, 1]$, $\delta$ is typically set to 1.5, and $\sigma$ is computed by

$$\sigma = \left(\frac{\Gamma(1+\delta) \times sin\left(\frac{\pi\delta}{2}\right)}{\Gamma(1+\delta) \times \delta \times 2^{\left(\frac{\delta-1}{2}\right)}}\right)^{\frac{1}{\delta}} \tag{33}$$

where the gamma function $\Gamma(x)$ is defined as $\Gamma(x) = (x-1)$.

Unlike other metaheuristic optimization algorithms, AVO features a more distinct exploration and exploitation mechanism. However, it still has some drawbacks, such as easily falling into locally optimal solutions and an imbalance between exploration and exploitation capabilities. To enhance AVO's performance, TAVO introduces three innovations. First, a tent chaotic map is utilized to initialize the population, ensuring diversity and preventing the algorithm from falling into local optima. Second, by fully leveraging historical optimal vulture information, the algorithm can achieve better solutions in the early stages. Third, a time-varying mechanism is designed to balance TAVO's exploration and exploitation abilities, enabling the algorithm to obtain superior solutions.

- *Tent chaotic mapping for population initialization*

Similar to other metaheuristic optimization algorithms, AVO uses randomly generated data for population initialization, which can hinder population diversity. Population diversity is crucial for the convergence speed and effectiveness of metaheuristic optimization algorithms, as it helps in finding the globally optimal solution more quickly. AVO's distinct exploration and exploitation mechanisms require effective guidance during the exploration stage; otherwise, the algorithm might fall into local optima during the exploitation stage. Therefore, it's essential for the population to cover the entire solution space as much as possible. However, random initialization in AVO often fails to achieve this, leading to local optimization issues.

Chaotic mapping, with its randomness and ergodicity, can maintain population diversity, helping the algorithm escape local traps and improve global exploration (*Rani, Jayan & Alatas, 2023*; *Gaddam et al., 2024*). *Kaur & Arora (2018)* demonstrated that the tent chaotic map performs best among ten chaotic maps in the whale optimization algorithm. The tent chaotic map generates a flatter and more uniform chaotic sequence compared to other maps. *Arora, Sharma & Anand (2020)* and *Zarei & Meybodi (2021)* also

found that the tent chaotic map significantly enhances algorithm performance in various optimization scenarios.

To address the aforementioned issues, TAVO employs tent chaotic mapping for population initialization. This approach ensures that the population evenly covers the entire solution space, enhancing algorithm performance during the exploration stage. The tent chaotic mapping is expressed by

$$x^{t+1} = tent(x^t) = \begin{cases} \dfrac{x^t}{u}, & 0 \le x < u \\ \dfrac{1 - x^t}{1 - u}, & u \le x \le 1 \end{cases}. \tag{34}$$

Based on prior research, tent chaotic mapping achieves optimal uniformity when $u = 0.5$ (*Carrasco-Olivera, Morales & Villavicencio, 2021*). Thus, to ensure a more evenly distributed sequence, this article employs $u = 0.5$. Consequently, Eq. (34) is substituted with

$$x^{t+1} = tent(x^t) = \begin{cases} 2x^t, & 0 \le x < 0.5 \\ 2(1 - x^t), & 0.5 \le x \le 1 \end{cases}. \tag{35}$$

Although tent chaotic mapping aims to achieve a uniform distribution, it still possesses some drawbacks. This is due to the limited byte length of computers, causing the value of $x$ to stabilize after a certain number of iterations. Two scenarios lead to tent chaotic mapping falling into a nonrandom cycle: when the initial value of $x$ is $\{0.2, 0.4, 0.6, 0.8\}$, and when the calculated value of $x$ becomes $\{0, 0.25, 0.5, 0.75\}$.

- *Individual history optimal solution*

From Eq. (22), during the exploration phase, if $p_1$ is greater than or equal to the random value $rand_{p1}^t$, the vulture relies solely on the current optimal vulture's information. While extensive exploration of unknown areas is essential, this can prevent early convergence, necessitating prolonged iteration to achieve the exploitation stage and better results, which is unsuitable for many real-time engineering problems. Excessive early-stage divergence can also lead to insufficient exploitation time later, causing the algorithm to fail and settle in a local optimum. To address this, TAVO records each vulture's historical optimal solution during the exploration stage and incorporates this in the location updating. This strategy curbs algorithm divergence early on and utilizes historical data to ensure the updated solution remains viable. Therefore, when the random number is less than or equal to parameter $p_1$ in the exploration stage, Eq. (22) is replaced by

$$D_i^t = \left| \omega_1^t \times C \times R_i^t + \omega_2^t \times C \times P_i - X_i^t \right| \tag{36}$$

where $P_i$ represents the best position the $i^{th}$ vulture has historically achieved, and $\omega_1^t$ and $\omega_2^t$ are values that vary with iterations, calculated using

$$\omega_1^t = 0.2 + \frac{1}{1.8 + e^{0.015 \times \left( \frac{T}{2} - t \right)}} \tag{37}$$

$$\omega_2^t = -\cfrac{1}{1.8 + e^{0.015 \times \left(\frac{T}{2} - t\right)}} - 0.8 \qquad (38)$$

respectively. Here, $T$ is the maximum number of iterations, and $t$ is the current iteration count. In Eq. (36), two parameters, $\omega_1^t$ and $\omega_2^t$, are added to control the influence of the current optimal vulture and the historical optimal vulture, respectively. This design ensures that even if $|F^t|$ remains greater than 1 in the algorithm's middle and late stages, the optimal vulture does not overly influence the current vultures, allowing for convergence. This time-varying mechanism is detailed in Eqs. (37) and (38).

- *Time-varying mechanism*

Equation (28) indicates that in the late exploitation stage, AVO treats the influence of the first and second groups of vultures on the current vulture equally during aggregation. This assumption is flawed. In the middle stages, the second group is necessary to boost exploration and avoid local optimization. However, in the later stages, equal influence from both groups hampers convergence due to similar exploration and exploitation abilities. To improve local exploitation in TAVO's later stages, two factors are introduced to control the influence of the first and second groups, with a time-varying mechanism to balance exploration and exploitation.

Thus, during the development stage, when $F_i^t$ is less than 0.5 and $rand_{p3}^t$ is greater than or equal to parameter $p_3$, the vulture's position update formula changes from Eq. (28) to

$$X_i^{t+1} = \frac{\omega_3^t \times A_{i1}^t + \omega_4^t \times A_{i2}^t}{2} \qquad (39)$$

where $\omega_3^t$ and $\omega_4^t$ vary with the number of iterations, calculated by

$$\omega_3^t = -0.2 \times e^{-2 \times \left(\frac{t}{T}\right)^2} - 0.6 \qquad (40)$$

$$\omega_4^t = 0.4 + 0.2 \times e^{-2 \times \left(\frac{t}{T}\right)^2} \qquad (41)$$

respectively. Here, $T$ is the maximum number of iterations, and $t$ is the current iteration count.

## REMD-LSTM-TAVO framework for wind speed forecasting

In this study, the components of the new hybrid modelling framework for short-term wind speed forecasting include: data pre-processing phase, construction of a new forecast model by combining REMD and LSTM algorithms, optimization of model parameters using TAVO, and model evaluation phase.

### Data preprocessing phase

The wind speed time series data are pre-processed before being used in the proposed model. This involves filling missing data with a weighted moving average (WMA) method (*Demirhan & Renwick, 2018*), smoothing the data with WMA filtering, and normalizing it using Z-score normalization (*Jain, Nandakumar & Ross, 2005*).

**Table 2 Configuration parameters for the REMD-LSTM-TAVO model.**

| | Parameter | Value/Range |
|---|---|---|
| *REMD* | *Intrinsic mode function* | 10 |
| *LSTM* | *Learning rate* | 0.001 |
| | *Number of layers* | 3 |
| | *Hidden units* | 200 |
| | *Batch size* | 64 |
| | *Dropout rate* | 0.2 |
| | *Epochs* | 150 |
| | *Activation functions* | Sigmoid for gate functions, $tanh(\cdot)$ for cell state updates. |
| *TAVO* | *Fitness function* | MSE |
| | *Population size (TAVO)* | 100 |
| | *Number of generations* | 500 |
| | *Crossover rate (TAVO)* | 0.7 |
| | *Mutation rate (TAVO)* | 0.01 |
| | *Encoding type (TAVO)* | Real-valued vectors |

### Construction phase of the REMD-LSTM combined wind speed forecasting model

At this phase, IMFs are derived using REMD decomposition to remove noise and stochastic volatility by decomposing and reconstructing the original series. Each IMF output is used to build a combined LSTM and REMD forecast model. The processed wind speed data are then used for optimization. The model is configured with the parameter values given in Table 2.

### Optimization phase

TAVO was chosen for its proven effectiveness in solving complex optimization problems, especially in scenarios that require a careful balance between exploration and exploitation over a large search space. Inspired by the social behaviour of vultures, this algorithm effectively avoids local optima and converges to the global optimum. In our study, TAVO is used to optimize the weighted coefficients of each IMF output to develop the most accurate forecasting model, using the mean squared error (MSE) as the fitness function for this optimization. Below is a detailed description of the TAVO configuration space, including the type of coding, the fitness function and other essential parameters.

- **Encoding type (representation scheme):** TAVO utilizes a real-valued vector encoding approach, where each individual (or vulture) is represented as a vector of continuous values. Each element of the vector corresponds to a specific parameter or coefficient that needs optimization. This encoding scheme is particularly suitable for refining the weighted coefficients of the IMFs generated through data decomposition, which are critical in enhancing the accuracy of wind speed forecasting models.

- **Population initialization:** The initial population of vultures is randomly generated within predefined bounds for each parameter. These bounds are chosen based on prior knowledge or exploratory runs to ensure that the initial solutions are diverse yet relevant to the problem domain. This diversity in the initial population helps the algorithm to explore the solution space effectively from the start.

- **Exploration and exploitation mechanism:** TAVO adapts its exploration and exploitation strategies dynamically throughout the optimization process. In the initial stages, the algorithm focuses on broad exploration to cover the search space extensively. As the process continues, the focus shifts towards exploitation, allowing the algorithm to fine-tune the solutions and approach the global optimum. The vultures' movement and decision-making process are modeled after real vultures' hunting strategies. Vultures tend to explore extensively when they are far from their target (global optimum) and exploit more aggressively as they get closer to it.

- **Selection mechanism:** A tournament selection process is used to choose individuals for reproduction. In this method, a subset of the population is randomly selected, and the best-performing individuals within this subset are chosen as parents for the next generation. This strategy maintains a balance between selecting high-quality candidates and preserving genetic diversity.

- **Crossover and mutation operators:** A linear crossover operator combines two parent solutions to produce offspring. This operator computes a weighted average of the parent vectors, facilitating smooth transitions between solutions and enabling the algorithm to explore new regions of the search space. An adaptive mutation strategy introduces small random perturbations to individual parameters. The size of these mutations decreases over time, reducing the likelihood of overshooting the optimal solution in the later stages of the optimization process.

- **Fitness function:** The fitness function used in TAVO is the MSE between the forecasted and actual wind speed values. This metric is chosen because it effectively quantifies the prediction accuracy of the model. The algorithm aims to minimize the MSE and thereby improve the accuracy of the forecast model. The fitness function ensures that only solutions that resulting in lower prediction errors are considered superior.

- **Convergence criteria:** The algorithm is set to terminate after a fixed number of generations unless the convergence criteria are met earlier. A stagnation detection mechanism is in place, where if no significant improvement in the best fitness value is observed over a certain number of generations, the algorithm concludes that it has reached convergence.

- **Diversity preservation:** To prevent premature convergence, TAVO incorporates a diversity maintenance mechanism. This is achieved by periodically introducing new random individuals or slightly modifying existing individuals that are too similar to others in the population. This approach helps to avoid local optima and ensures a more robust search process.

**Table 3 Performance metrics for wind speed forecasting model evaluation.**

| Metric | Description | Formula |
|--------|-------------|---------|
| MAE | Mean absolute error | $MAE = \frac{1}{N}\sum_{i=1}^{N}\left\vert p_{act}^{i} - p_{pred}^{i}\right\vert$ |
| MSE | Mean square error | $MSE = \frac{1}{N}\sum_{i=1}^{N}\left(p_{act}^{i} - p_{pred}^{i}\right)^{2}$ |
| RMSE | Root mean square error | $RMSE = \sqrt{\frac{1}{N}\sum_{i=1}^{N}\left(p_{act}^{i} - p_{pred}^{i}\right)^{2}}$ |
| R | Pearson correlation coefficient | $R = \dfrac{\sum_{i=1}^{N}\left(p_{act}^{i} - \overline{p_{act}^{i}}\right)\left(p_{pred}^{i} - \overline{p_{pred}^{i}}\right)}{\sqrt{\sum_{i=1}^{N}\left(p_{act}^{i} - \overline{p_{act}^{i}}\right)^{2}}\sqrt{\sum_{i=1}^{N}\left(p_{pred}^{i} - \overline{p_{pred}^{i}}\right)^{2}}}$ |

This configuration space of TAVO highlights the algorithm's adaptability and effectiveness in optimizing complex forecasting models. Its ability to dynamically balance exploration and exploitation, along with its sophisticated selection and mutation strategies, makes it a powerful tool for achieving high-precision wind speed predictions.

*Model evaluation phase*

The performance of the model is evaluated using mean absolute error (MAE), MSE, and root mean square error (RMSE). The PCC, $R$ measures the strength and direction of the linear relationship between two variables. The formulas for these performance metrics are provided in Table 3. Here, $N$ denotes the length of the time series, $p_{act}^{i}$ represents the actual wind speed time series, $p_{pred}^{i}$ denotes the predicted wind speed time series, $\overline{p_{act}^{i}}$ is the mean value of the actual wind speed time series, and $\overline{p_{pred}^{i}}$ is the mean value of the predicted wind speed time series. The $R$ value ranges between 0 and 1. As the $R$ value approaches 1, the model's prediction performance improves. The closer the MAE, MSE, and RMSE values are to 0, the lower the prediction error of the model.

# RESULTS AND DISCUSSION

In this section, we present a study conducted using wind speed data collected from the Bandırma meteorological station in the Marmara region, one of Turkey's prime areas for wind energy potential. We present the proposed REMD-LSTM-TAVO wind speed forecasting model and compare its performance with single LSTM, EMD-LSTM, REMD-LSTM, and REMD-LSTM-CPSO models. We also investigate the influence of REMD and TAVO on the model performance.

## Dataset

The dataset was obtained by averaging 10-h data collected from the station shown in Fig. 2. Wind speed and direction measurements spanned from 2008 to 2014, totaling 6 and 2/3 years. Approximately 51,725 h of wind speed data were collected from the Bandırma station. The Bandırma station, chosen arbitrarily, is isolated from other measurement locations and is not near the Çanakkale and Bosphorus Straits. Uncertainty analysis

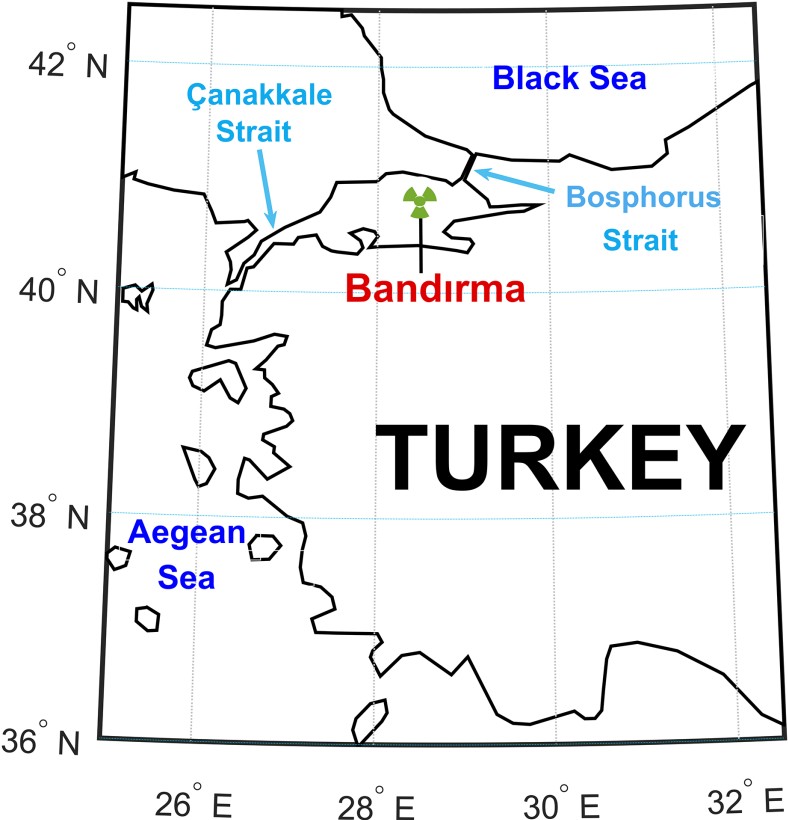

**Figure 2 The location of the wind station where the data used in the study were collected.**

validated the experimental data, establishing upper and lower confidence levels for forecasting values (*Akçay & Filik, 2017*).

The dataset used in the study consists of 5,173 wind speed records, ranging from 0.6 to 25.7 m/s. The mean wind speed in the dataset is 3.99 m/s, with a standard deviation of 2.51 m/s. For model training, wind speed values between 0.7 and 25.7 m/s were used, with a mean of 4.04 m/s and a standard deviation of 2.54 m/s. Model performance was evaluated using a test dataset with wind speeds ranging from 0.6 to 11.6 m/s. The mean wind speed in the test dataset is 3.71 m/s, with a standard deviation of 2.29 m/s.

To demonstrate the independence of model performance from the data, the dataset was partitioned into three independent segments of 70%, 15% and 15% for training, validation and testing respectively. The first 3,623 wind speed data points were used for model training, followed by validation using the subsequent 775 data points. Finally, model performance was evaluated using the last 775 wind speed data points in the dataset.

The models were trained and tested on a personal computer equipped with an Intel Core i7-10875H processor, an 8 GB NVIDIA RTX 3070 graphics card and 16 GB RAM. Code for all models was compiled in the MATLAB 2021b environment.

## Experimental results and discussions

The aim of this study is to develop a high-performance wind speed forecasting model capable of addressing for the non-linear dynamics inherent in wind speed time series. To achieve this, the wind speed time series data collected from the Bandırma meteorological station were first decomposed into sub-bands using EMD and REMD techniques. The impact of these decomposed bands on model performance was analyzed to construct a robust short-term wind speed forecasting model. Subsequently, an artificial intelligence-based LSTM network was trained using the decomposed sub-bands.

To further improve the predictive performance of the model and to address the issue of hyperparameter optimization within the deep learning network, optimization techniques were applied. In this study, CPSO and TAVO were used as the optimization algorithms. While we have compared TAVO with other widely used optimization methods, we have presented the results of CPSO for the sake of clarity and relevance. CPSO is a well-known benchmark in the field, recognized for its robustness on tasks similar to ours, making it an appropriate choice for comparison. The effects of these optimization algorithms on the performance of the models were systematically evaluated.

In TAVO, constraints are managed through boundary handling and penalty functions. For decision variable bounds, TAVO uses methods such as clamping or reflecting to ensure that the variables remain within the specified bounds. When dealing with constraint functions, TAVO applies penalty functions to the objective function for any constraint violations, thus discouraging the exploration of infeasible solutions. In CPSO, boundary constraints are handled by clamping, which adjusts variables to the nearest feasible boundary if they exceed the limits. Similarly, CPSO deals with constraint functions by incorporating penalty functions into the fitness function to penalize constraint violations, thereby guiding the search towards feasible solutions. These approaches ensure that both TAVO and CPSO effectively navigate constraints during the optimization process.

The performance of all models developed in this study was rigorously measured using the error metrics described in "Model Evaluation Phase". This comprehensive approach ensures that the resulting wind speed forecasting model is both effective in dealing with non-linear dynamics and optimized for high forecasting accuracy.

To address the nonlinear dynamics in the wind speed time series used in this study, a dataset of 5,173 wind speed values was decomposed into 10 sub-bands using the EMD technique, as shown in Fig. 3. From these decomposed IMFs, 10 hybrid wind speed forecasting models were created, each excluding one specific IMF to analyze its impact on model performance. The test phase utilized a subset of 775 data points, representing 15% of the dataset, marked by a red line in Fig. 3.

The performance of all models was evaluated using MAE, MSE, RMSE, and $R$ error metrics. To assess the effect of each IMF on the hybrid model, the forecasting performance of the models excluding each IMF was compared with a single LSTM model and a hybrid model including all IMFs. The single LSTM model served to measure the improvement in forecasting performance due to the decomposition method.

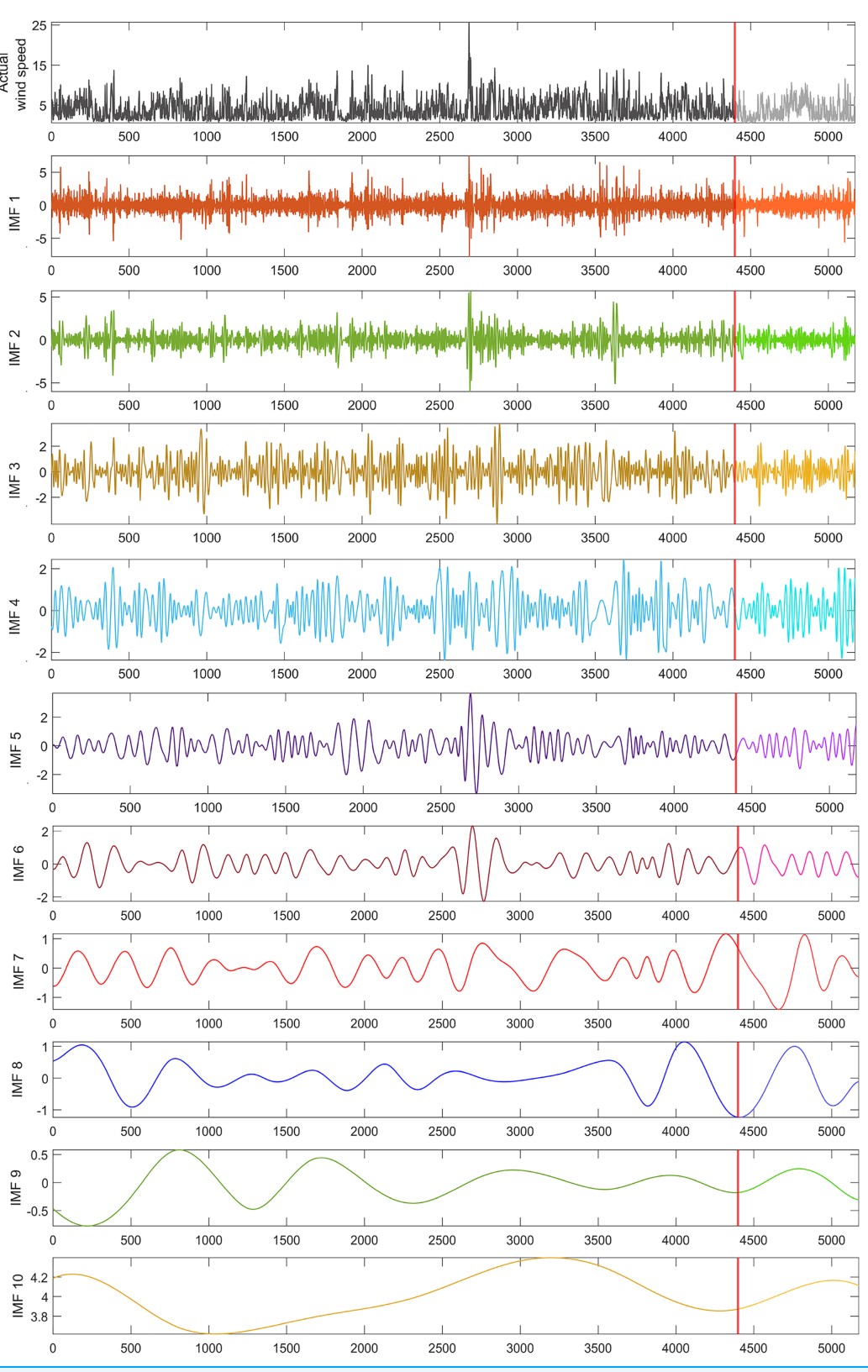

**Figure 3 EMD sub-bands for the wind speed time series used in the study.**

**Table 4 Comparison of the average performance metrics for the EMD-LSTM wind speed forecasting models.**

| Station | Model | MAE | MSE | RMSE | $R$ |
|---------|-------|-----|-----|------|-----|
| *Bandırma* | LSTM | 1.3844 | 3.2063 | 1.7906 | 0.5688 |
| | EMD-LSTM | 0.7663 | 1.1902 | 1.0910 | 0.8696 |
| | EMD (except IMF1)-LSTM | 0.8941 | 1.3370 | 1.1597 | 0.7836 |
| | EMD (except IMF2)-LSTM | 1.1935 | 2.7030 | 1.6441 | 0.7016 |
| | EMD (except IMF3)-LSTM | 1.1079 | 2.2629 | 1.5043 | 0.7414 |
| | EMD (except IMF4)-LSTM | 1.0549 | 2.1405 | 1.4630 | 0.7539 |
| | EMD (except IMF5)-LSTM | 0.9717 | 1.6436 | 1.2820 | 0.7596 |
| | EMD (except IMF6)-LSTM | 1.0503 | 2.3338 | 1.5277 | 0.7541 |
| | EMD (except IMF7)-LSTM | 1.1680 | 2.5293 | 1.5904 | 0.7225 |
| | EMD (except IMF8)-LSTM | 0.8129 | 1.3939 | 1.1806 | 0.8321 |
| | EMD (except IMF9)-LSTM | 0.8484 | 1.4903 | 1.2208 | 0.8019 |
| | EMD (except IMF10)-LSTM | 0.8034 | 1.3355 | 1.1556 | 0.8599 |

Ten sub-LSTM models were developed by feeding each IMF into the LSTM network, and these sub-models were combined to form the EMD-LSTM wind speed model, which included all sub-bands. The EMD-LSTM model was compared with the single LSTM model to determine the effect of including all sub-bands on the short-term wind speed forecasting performance. The EMD-LSTM model achieved forecasting performance values of 0.7663, 1.1902, 1.0910, and 0.8696 for MAE, MSE, RMSE, and $R$ respectively, while the corresponding values for the single LSTM model were 1.3844, 3.2063, 1.7906, and 0.5688.

Comparing the EMD-LSTM model to the single LSTM model revealed improvements of 44.64%, 62.87%, 39.07%, and 30.08% in MAE, MSE, RMSE, and $R$, respectively. To further investigate the impact of each IMF, AI-based wind speed forecasting models were built excluding each IMF. The forecasting performances of these models, based on MAE, MSE, RMSE, and $R$ are detailed in Table 4.

Table 4 indicates that the highest performance was achieved by the model excluding IMF10, which predicted actual wind speeds with errors of 4.61%, 10.87%, 5.59%, 0.97% in MAE, MSE, RMSE, and $R$, respectively, compared to the EMD-LSTM model including all IMFs. The lowest performance was observed in the model excluding IMF2, forecasting wind speeds 35.79%, 55.96%, 33.64%, and 16.80% worse in MAE, MSE, RMSE, $R$, respectively, than the EMD-LSTM model. These results demonstrate that the EMD decomposition technique improves short-term wind speed forecasting performance by at least 39%.

To address the impact of nonlinear dynamics in the wind speed time series utilized in this study, a dataset of 5,173 wind speed measurements was decomposed into 10 sub-bands using the REMD technique, as illustrated in Fig. 4. From these decomposed IMFs, 10 hybrid forecasting models were developed, each excluding one specific IMF to evaluate its influence on model performance. To ensure that the performance of the models was independent of the dataset, it was divided into 70% for training, 15% for validation, and

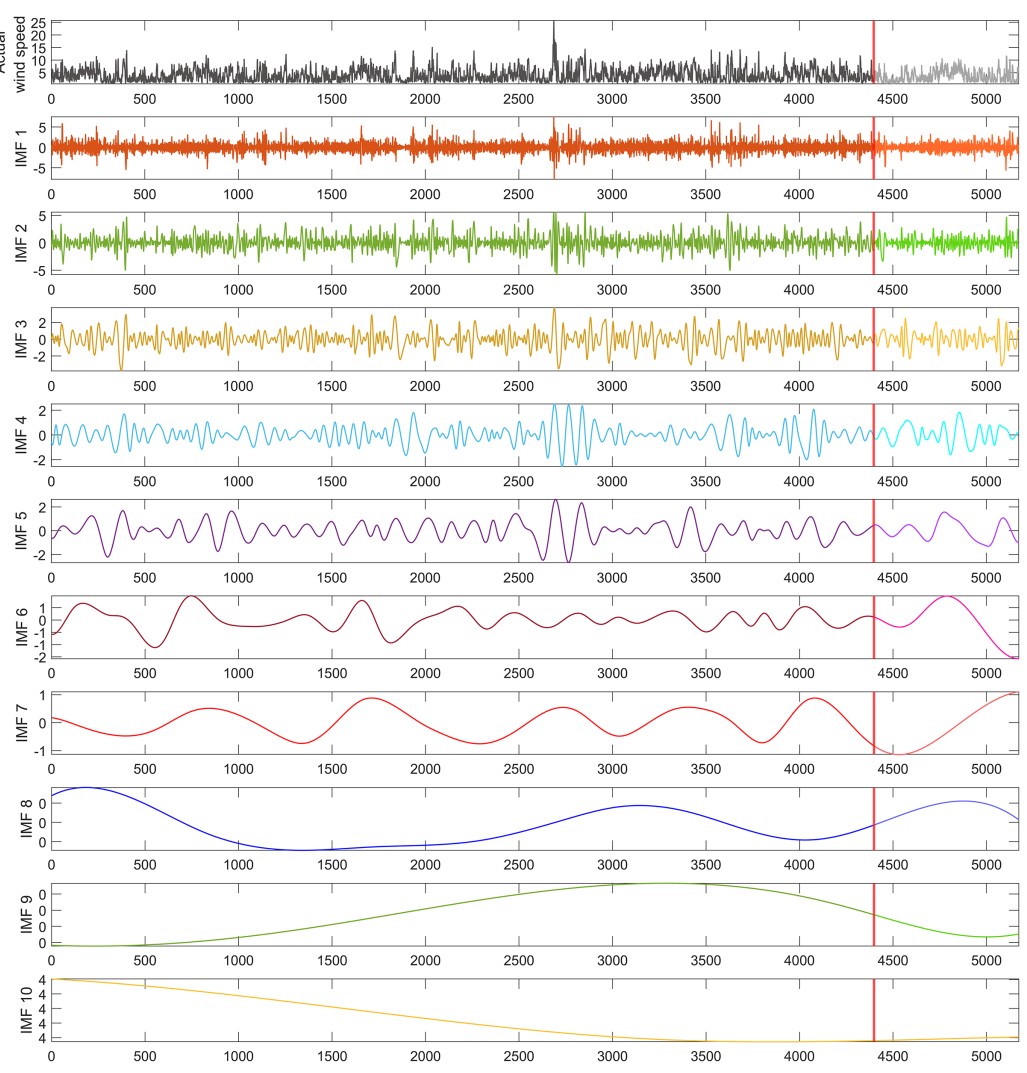

**Figure 4** REMD sub-bands for the wind speed time series used in the study.

15% for testing. The segment marked by the red line in Fig. 4, containing 775 data points, was allocated for the model's testing phase, representing 15% of the wind speed time series dataset. This approach allowed for a comprehensive evaluation of the models' predictive accuracy and robustness across different stages of the data.

Ten sub-LSTM models were developed by feeding each IMF individually into the LSTM network. These sub-models were then aggregated to form the REMD-LSTM wind speed model, which included all sub-bands. The REMD-LSTM model was compared with the single LSTM model to assess the effect of including all sub-bands on the short-term wind speed forecasting performance. The REMD-LSTM model achieved forecasting performance values of 0.7354 for MAE, 1.0516 for MSE, 1.0255 for RMSE, and 0.8925 for R. In comparison, the single LSTM model yielded values of 1.3844 for MAE, 3.2063 for MSE, 1.7906 for RMSE, and 0.5688 for R. These results underline the superior forecasting

**Table 5 Comparison of the average performance metrics for the REMD-LSTM wind speed forecasting models.**

| Station | Model | MAE | MSE | RMSE | R |
|---------|-------|-----|-----|------|---|
| *Bandırma* | LSTM | 1.3844 | 3.2063 | 1.7906 | 0.5688 |
| | REMD-LSTM | 0.7354 | 1.0516 | 1.0255 | 0.8925 |
| | REMD (except IMF1)-LSTM | 0.8930 | 1.3252 | 1.1512 | 0.7893 |
| | REMD (except IMF2)-LSTM | 1.2031 | 2.6243 | 1.6138 | 0.7399 |
| | REMD (except IMF3)-LSTM | 1.1080 | 2.1265 | 1.4583 | 0.7501 |
| | REMD (except IMF4)-LSTM | 0.9836 | 2.6174 | 1.2718 | 0.7726 |
| | REMD (except IMF5)-LSTM | 0.9699 | 1.6048 | 1.2668 | 0.7845 |
| | REMD (except IMF6)-LSTM | 0.9784 | 1.7335 | 1.3166 | 0.7782 |
| | REMD (except IMF7)-LSTM | 1.0664 | 1.8950 | 1.3766 | 0.7627 |
| | REMD (except IMF8)-LSTM | 0.7807 | 1.2265 | 1.1075 | 0.8528 |
| | REMD (except IMF9)-LSTM | 0.8027 | 1.2456 | 1.1161 | 0.8335 |
| | REMD (except IMF10)-LSTM | 0.7665 | 1.1932 | 1.0923 | 0.8621 |

capability of the REMD-LSTM model, highlighting its improved accuracy and reliability in wind speed forecasting.

A comparison of the REMD-LSTM model with the single LSTM model showed significant improvements in forecasting performance: 46.87% in MAE, 67.20% in MSE, 42.72% in RMSE, and 32.37% in the *R*. To further investigate the impact of each IMF, AI-based wind speed forecasting models were developed, each excluding one IMF. The forecasting performance of these models, evaluated using MAE, MSE, RMSE, and R, is detailed in Table 5.

Table 5 shows that the highest performance was achieved by the model excluding IMF10, which predicted actual wind speeds with errors of 4.05%, 11.86%, 6.11%, and 3.04% in MAE, MSE, RMSE, and R, respectively, compared to the REMD-LSTM model that included all IMFs. Conversely, the lowest performance was observed in the model excluding IMF2, forecasting wind speeds with 61.12%, 40.07%, 63.54%, and 15.26% worse in MAE, MSE, RMSE, and R, respectively, than the REMD-LSTM model. These findings indicate that the REMD decomposition technique improves short-term wind speed forecasting performance by at least 42.50%.

The short-term wind speed forecasting results for the single LSTM model, the EMD-LSTM model (which incorporates all IMFs), and the REMD-LSTM model (also incorporating all IMFs) are illustrated in Fig. 5. The REMD-LSTM model shows a significant improvement in forecast performance compared to both the EMD-LSTM and single LSTM models, particularly in its accuracy in predicting instantaneous wind speed fluctuations. This improvement highlights the superior ability of the REMD-LSTM model to effectively capture and respond to the dynamic variations inherent in wind speed time series data. When comparing the prediction performance of the EMD-LSTM and REMD-LSTM models with the single LSTM model, the EMD-LSTM and REMD-LSTM models demonstrate at least 39% and 42.50% better forecasting performance, respectively.

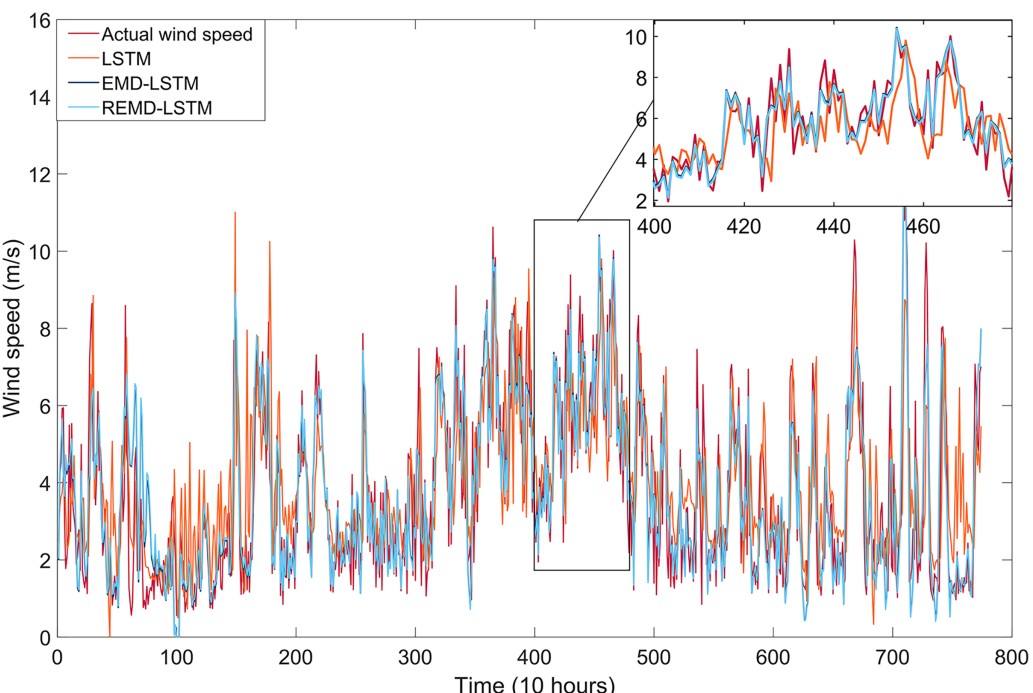

**Figure 5 Wind speed forecasting results of models built with single LSTM, EMD-LSTM, and REMD-LSTM.**

Furthermore, when the REMD-LSTM and EMD-LSTM models are evaluated using the MAE, MSE, RMSE, and R performance metrics, the REMD-LSTM model exhibits improvements of 4.03%, 11.64%, 6.00%, and 2.29%, respectively, over the EMD-LSTM model.

The results presented in the study have confirmed that the forecasting model constructed using REMD outperformed the one constructed using EMD in forecasting actual wind speeds. The focus of the study then shifted to improving the performance of the artificial intelligence-based wind speed forecasting model developed using the REMD technique. A natural meta-heuristic optimization algorithm based on tent chaotic mapping, known as TAVO, was employed to address both the hyper-parameter problem within the model and to improve forecast accuracy. The impact of the chosen optimization algorithm on the forecasting performance of the model was compared with the results obtained using the CPSO algorithm. For both the CPSO and TAVO algorithms utilized in the study, the fitness function was set as MSE. When MSE was set as the fitness function, the regression curves covering the training, validation, and test phases of the REMD-LSTM-CPSO and REMD-LSTM-TAVO hybrid wind speed forecasting models are shown in Figs. 6 and 7, respectively. The forecasting performances of the REMD-LSTM-CPSO and REMD-LSTM-TAVO models on the test data were measured to be 91.49% and 93.96%, respectively.

The short-term wind speed forecasting results for the REMD-LSTM-CPSO and REMD-LSTM-TAVO hybrid models developed in this study are shown in Fig. 8. It has been tested that the REMD-LSTM-TAVO model is more robust to the volatility of the actual wind

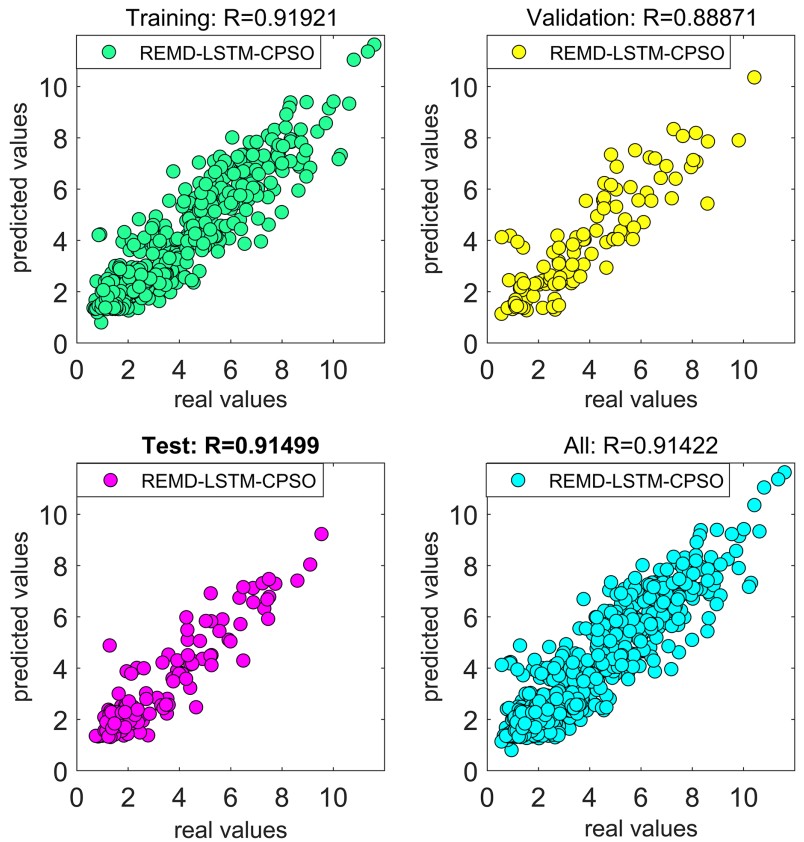

**Figure 6 Regression curve of the REMD-LSTM-CPSO model.**

speed compared to the REMD-LSTM-CPSO model. The REMD-LSTM-TAVO model shows a significant improvement in forecast performance, particularly in its accuracy in forecasting instantaneous wind speed fluctuations. This improvement highlights the superior ability of the REMD-LSTM-TAVO model to effectively capture and respond to the dynamic variations inherent in wind speed time series data.

The performance for all models developed in the study are presented in Table 6, evaluated based on MAE, MSE, RMSE, and R. The proposed REMD-LSTM-TAVO hybrid model exhibits forecast performance values of 0.6785 for MAE, 0.8692 for MSE, 0.9323 for RMSE, and 0.9396 for $R$. Based on the R performance metric, the proposed REMD-LSTM-TAVO hybrid model shows improved forecasting performance, with improvements of 37.08%, 7.00%, 4.71%, and 2.47% over the LSTM, EMD-LSTM, REMD-LSTM, and REMD-LSTM-CPSO models, respectively.

In the context of evaluating the performance of the proposed REMD-LSTM-TAVO hybrid model, it is essential to consider the time complexity, especially in comparison with the CPSO algorithm. The AVOA, when augmented with tent chaotic mapping and a time-varying mechanism, has a time complexity that depends mainly on the population size ($N$) and the number of iterations ($T$). This gives the TAVO a time complexity of $O(N \times T)$. The integration of chaotic mapping enhances the exploration capabilities, while the time-

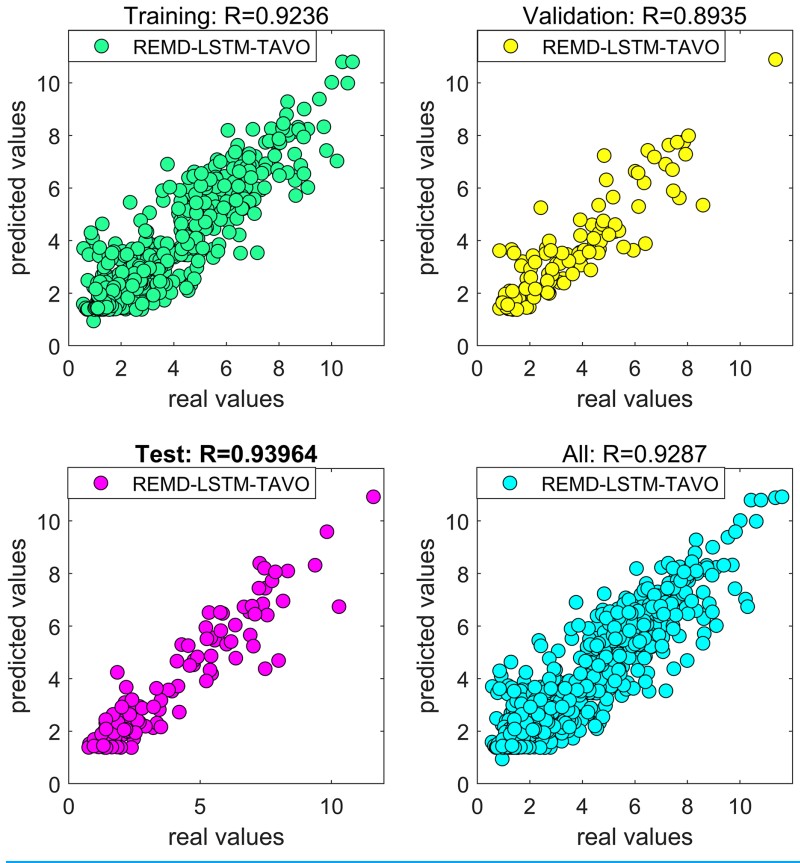

**Figure 7 Regression curve of the REMD-LSTM-TAVO model.**

varying mechanism provides a better balance between exploration and exploitation phases, leading to improved convergence speed and accuracy. On the other hand, the CPSO algorithm, which also relies on the population size and the number of iterations for its time complexity, has a similar time complexity of $O(N \times T)$. CPSO uses chaotic sequences to avoid premature convergence and to enhance the global search capability of the standard PSO algorithm. When comparing the two, both algorithms aim to improve the optimization process by integrating chaotic mechanisms. However, the TAVO's use of a time-varying mechanism provides an additional layer of dynamic adjustment throughout the optimization process. This mechanism helps to fine-tune the search process over time, potentially offering better performance in terms of convergence speed and solution accuracy compared to CPSO.

In practical terms, while both algorithms offer efficient computational performance, the TAVO-enhanced AVO's ability to dynamically adjust its parameters during the optimization process could result in more robust and accurate short-term wind speed forecasts, as reflected in the empirical results presented in this study. Thus, the REMD-LSTM-TAVO hybrid model, therefore, not only benefits from an improved forecast performance but also maintains computational efficiency comparable to that of the CPSO algorithm, making it a viable option for real-time applications.

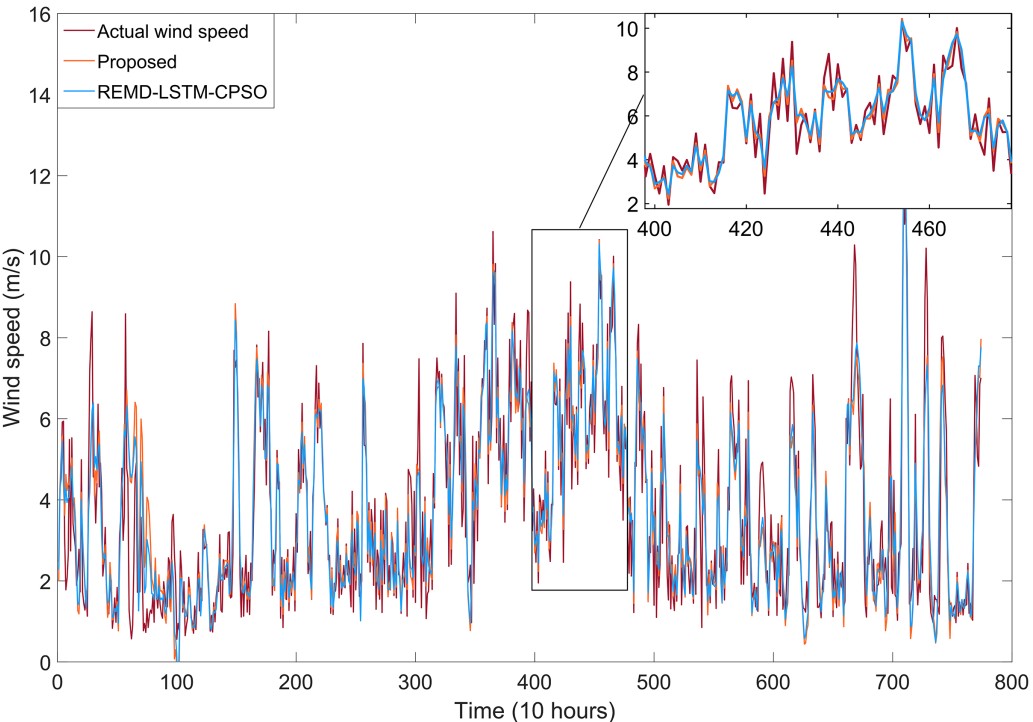

**Figure 8 Wind speed forecasting results for models developed with REMD-LSTM-CPSO and the proposed REMD-LSTM-TAVO.**

**Table 6 Comparison of the performance of the wind speed forecasting models according to the MAE, MSE, RMSE and performance metrics.**

| Model | MAE | MSE | RMSE | R (Test) |
|---|---|---|---|---|
| LSTM | 1.3844 | 3.2063 | 1.7906 | 0.5688 |
| EMD-LSTM | 0.7663 | 1.1902 | 1.0910 | 0.8696 |
| REMD-LSTM | 0.7354 | 1.0516 | 1.0255 | 0.8925 |
| REMD-LSTM-CPSO | 0.7023 | 0.9013 | 0.9494 | 0.9149 |
| REMD-LSTM-TAVO | 0.6785 | 0.8692 | 0.9323 | 0.9396 |

# CONCLUSIONS AND FUTURE RESEARCH

High-precision wind speed forecasts are essential for the efficient development and use of wind energy. Uncertainty and non-stationarity of wind speed suggest the possibility of using combined models for high-precision and reliable wind speed forecasting. In this article, a hybrid model called REMD-LSTM-TAVO based on LSTM network and REMD decomposition method with TAVO is developed for short term wind speed forecasting. The REMD-LSTM-TAVO hybrid model developed in this study marks a significant advance in short-term wind speed forecasting by effectively dealing with the non-linear and non-stationary characteristics of wind speed data. By decomposing wind speed data into IMFs using REMD and integrating them with an LSTM network, the model significantly improves forecast accuracy. The TAVO further optimizes the

hyperparameters of the model, resulting in superior performance compared to the CPSO algorithm.

The empirical results indicate that the REMD-LSTM-TAVO model significantly outperforms other models, demonstrating notable improvements in the performance metrics such as MAE, MSE, RMSE, and *R*. In particular, the model achieved a 37.08% improvement in R over the standard LSTM model, highlighting its robustness and reliability. This high level of accuracy plays a key role in optimizing the integration of wind energy into the grid, thereby improving energy management and sustainability efforts.

The study also highlights the practical implications of the REMD-LSTM-TAVO hybrid model. Its computational efficiency, comparable to the CPSO algorithm, ensures its feasibility for real-time applications. The dynamic adaptation capabilities of the TAVO algorithm provide an additional advantage, making the model adaptable to varying wind speed patterns and more effective in capturing the dynamic variations inherent in wind speed data. This capability is critical for optimizing the integration of wind energy into the grid, thereby improving energy management and sustainability.

In future studies, the integration of this model into operational wind energy management systems can further improve the efficiency and reliability of renewable energy sources. This integration will play a crucial role in advancing the sustainability and effectiveness of wind energy utilization.

## NOMENCLATURE

| | |
|---|---|
| **ANN** | artificial neural network |
| **ARIMA** | autoregressive integrated moving average |
| **AVO** | African vultures optimization |
| **BiGRU** | bi-directional gated recurrent unit |
| **BPNN** | back propagation neural network |
| **CEEMDAN** | complete ensemble empirical mode decomposition with adaptive noise |
| **CNN** | convolutional neural network |
| **ConvLSTM** | convolutional long short-term memory |
| **EA** | evolutionary algorithms |
| **ELM** | extreme learning machine |
| **EMD** | empirical mode decomposition |
| **EEMD** | ensemble empirical mode decomposition |
| **ENN** | Elman neural network |
| **EO** | extremal optimization |
| **FE** | fuzzy entropy |
| **FL** | fuzzy logic |
| **GRU** | gated recurrent unit |
| **GWO** | grey wolf optimizer |
| **HMD** | hybrid mode decomposition |
| **HTSD** | hybrid time series decomposition |

| | |
|---|---|
| **ICEEMDAN** | improved complete ensemble empirical mode decomposition with adaptive noise |
| **IChOA** | improved chimp optimization algorithm |
| **IMF** | intrinsic mode function |
| **LSTM** | long short-term memory |
| **MAE** | mean absolute error |
| **MIC** | maximum information coefficient |
| **MLP** | multi-layer perceptron |
| **MOBBSA** | multi-objective binary backtracking search algorithm |
| **MOO** | multi-objective optimization |
| **MOGWO** | multi-objective grey wolf optimization |
| **MOPSO** | multi-objective particle swarm optimization |
| **MSE** | mean squared error |
| **NSGA-II** | non-dominated sorting genetic algorithm II |
| **NWP** | numerical weather prediction |
| **PACF** | partial autocorrelation function |
| **PCC** | Pearson correlation coefficient |
| **RBF** | radial basis function |
| **REMD** | robust empirical mode decomposition |
| **RF** | random forest |
| **RFE** | recursive feature elimination |
| **RMSE** | root mean square error |
| **RNN** | recurrent neural network |
| **Seq2Seq** | sequence-to-sequence |
| **SSA** | singular spectrum analysis |
| **SVRM** | support vector regression machine |
| **TAVO** | improved African vulture optimization algorithm based on tent chaotic mapping and time-varying mechanism |
| **TVFEMD** | time varying filtering-based empirical mode decomposition |
| **VMD** | variational mode decomposition |
| **WMA** | weighted moving average |
| **WNN** | wavelet neural network |
| **XGBoost** | extreme gradient boosting |

### Funding

The authors received no funding for this work.

### Competing Interests

The authors declare that they have no competing interests.

## Author Contributions

- Caner Barış conceived and designed the experiments, performed the experiments, analyzed the data, performed the computation work, prepared figures and/or tables, and approved the final draft.
- Cağfer Yanarateş conceived and designed the experiments, performed the experiments, prepared figures and/or tables, authored or reviewed drafts of the article, and approved the final draft.
- Aytaç Altan conceived and designed the experiments, performed the experiments, analyzed the data, performed the computation work, prepared figures and/or tables, authored or reviewed drafts of the article, and approved the final draft.

## Data Availability

The raw data are available in the Supplemental File.

## Supplemental Information

Supplemental information for this article can be found online at http://dx.doi.org/10.7717/peerj-cs.2393#supplemental-information.

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
