# Peer review of "A robust chaos-inspired artificial intelligence model for dealing with nonlinear dynamics in wind speed forecasting"

_PeerJ Computer Science, doi:10.7717/peerj-cs.2393_

## Round 0.1 · original submission · Major Revisions

Dear authors,

Thank you for submitting your article. Reviewers have now commented on your article and suggest major revisions. When submitting the revised version of your article, it will be best to address the following:

1. There is not a clear categorization of related works. It is suggested to use a table to summarize the overall information of the relevant literature and to present the main features of the paper. A summary table of related work and main contributions should be better highlighted. This table can be helpful to clarify what differences there are between the proposed approach and previous works.
2. In general, the literature review is not sufficient. It is more of the type “researcher X did Y” rather than an authoritative synthesis assessing the current state-of-the-art. Where do we stand today? What approaches are there in the literature to model the problem? What are the main differences between them? What are their weaknesses and strengths? Please evaluate that how this study is different from others in the related work section? What do you have where others do not? Why you are better or how? What’s new/novel here?
3. Integrating chaos into optimization is routine work. More recent works on this topic is not deeply explored. It seems as if this is a novel approach in the paper, in fact it is not. To increase professionalism, references should be reinforced with recent papers from reputable journals. Recent literature should be deeply analyzed.
4. Some paragraphs are too long to read. You should try for readability and comprehensibility. Some paragraphs may be divided into two or three.
5. The reason for selecting the african vultures optimization is not discussed. There is no detailed discussion and the you do not explain why they used especially this search method among many others. Furthermore, why only chaotic particle swarm optimization is selected for comparison is not presented.
6. Configuration space of evolutionary algorithm should be detailed. It should be more specific and comprehensive. Encoding type (representation scheme) and fitness function should be provided.
7. A table with parameter setting for experimental results and analysis should be included in order to clearly describe them.
8. How the constraints are coped with in african vultures optimization and particle swarm optimization is not clear. Constraint handling methods for decision variables boundaries and constraint functions are not described.
9. Equations should be used with equation number. Please do not use “following”, “as follows”, etc. Explanation of the equations should be checked. All variables should be written in italic as in the equations. Their definitions and boundaries should be explained. Relevant references should be given for the equations.
10. Additional comments about the reached results should be included. Graphics and charts need more explanation.
11. The paper lacks the running environment, including software and hardware. The analysis and configurations of experiments should be presented in detail for reproducibility. It is convenient for other researchers to redo your experiments and this makes your work easy acceptance. A table with parameter settings for experimental results and analysis should be included in order to clearly describe them.
12. You should clarify the pros and cons of the methods. What are the limitation(s) methodology(ies) adopted in this work? Please indicate practical advantages, and discuss research limitations.

Best wishes,

Reviewer 1 ·

Basic reporting

.

Experimental design

.

Validity of the findings

.

Additional comments

Overall, the study does not make a significant contribution to the existing body of knowledge. Additionally, it lacks novel findings, and the results are unremarkable. Furthermore, the quality of the figures is very poor.

Reviewer 2 ·

Basic reporting

The authors focus on developing a robust wind speed forecasting model capable of handling non-linear dynamics to minimize losses and improve wind energy efficiency. The data were decomposed into IMFs using REMD. The extracted IMFs were then fed into an LSTM, with its parameters estimated using the AVO algorithm. The study emphasizes the potential of utilizing advanced optimization techniques and deep learning models to enhance wind speed forecasting, ultimately contributing to more efficient and sustainable wind energy generation. The authors should improve the manuscript in the following areas:

State the main theme at the beginning of the article for improved clarity.
Restructuring paragraphs in Introduction section to enhance overall logical flow.
Compare with existing research to bolster the progressiveness of their arguments.
Provide more specific implications in the conclusion to help readers better understand the practical significance of the research.
Point out potential directions for further research in the conclusion.

Experimental design

see the basic reporting

Validity of the findings

see the basic reporting

Additional comments

see the basic reporting

Reviewer 3 ·

Basic reporting

The paper is generally clear and professionally written, with unambiguous and well-structured English used throughout. The introduction provides a thorough context for the study, highlighting the importance of accurate wind speed forecasting in optimizing wind energy utilization. The background is well-supported with relevant literature references, setting the stage for the motivation behind the research. The structure of the paper conforms to PeerJ standards and improves clarity by logically presenting the methodology, results, and discussions.

However, some specific areas could benefit from enhancement:

The explanation of the African Vultures Optimization (AVO) algorithm and its chaotic mapping mechanism needs more detailed description to aid readers unfamiliar with these techniques.
Definitions and terminologies related to the proposed methodologies should be explicitly stated for clarity.

Suggested Improvements:

Enhance the description of the AVO algorithm and tent chaotic mapping in the methodology section.
Explicitly define all terms and acronyms used throughout the paper.
Cite additional recent literature to provide a more comprehensive background.

Experimental design

The experimental design is within the aims and scope of the journal, and the investigation is conducted to a high technical standard. The methodology is well-detailed, allowing for replication. The data preprocessing steps, including filling missing data with a weighted moving average (WMA) method and normalizing data using Z-score normalization, are clearly described.

However, there are areas where the paper could be improved:

A more detailed explanation of the feature selection process and the rationale behind choosing specific features would enhance the comprehensiveness of the study.
The evaluation methods and assessment metrics are adequately described, but further details on the computing infrastructure and reproduction scripts would be beneficial.
Suggested Improvements:

Provide a more detailed explanation of the feature selection process and its significance.
Include more information about the computing infrastructure and reproduction scripts to ensure full replicability.

Validity of the findings

Comment:
The findings are well-stated and supported by the results. The experiments and evaluations are performed satisfactorily, demonstrating the proposed model's effectiveness in improving wind speed forecasting accuracy. The results are clearly presented and discussed, showing significant improvements over existing methods.

However, the paper could benefit from a deeper discussion of the implications of the findings and any potential limitations:

Discuss the limitations of the proposed model and the challenges encountered during the research.
Suggest future research directions to address unresolved questions.
Suggested Improvements:

Provide a detailed discussion on the limitations of the proposed model.
Outline future research directions to explore other optimization algorithms and different geographical regions.

Additional comments

General Comments:

The paper presents a promising approach to improving wind speed forecasting through the integration of robust empirical mode decomposition, long short-term memory neural networks, and optimization algorithms.
The comprehensive experimental validation and comparison with existing methods are commendable.
With enhancements in the explanation of algorithms, feature selection rationale, and practical implications, the paper can make a significant contribution to the field of renewable energy forecasting.
Overall, the paper is well-structured and presents a thorough investigation into an important area of research. Addressing the suggested improvements will enhance the clarity and impact of the study.

Reviewer 4 ·

Basic reporting

1. In the introduction part, the contributions of this article should be rewritten more clearly. The authors could summarize and list some contribution points.
2.It would be better to more the Nomenclature into the beginning of this paper.
3.The authors are invited to perform a thorough proofread of their manuscript, as I can still spot some spelling/grammar mistakes in the article
4. The disadvantage or the limitation of the proposed method must be described in conclusion
5.Suggest to add recent literature work in the introduction.

Experimental design

1. Abstract should be improved. Explain a problem and its proposed solution and benefits over existing strategies
2. Introduction is short and not complete. My suggestion is to divide the introduction into three subsections: 1) motivation and incitement, 2) literature review and 3) contribution and paper organization

3.Figure quality can be improved for Fig 3,4 and 8
4. Clearly mention wny REMD is choosen over other data decompostion techinque like CEEMDAN,VMD etc.Give a comparitive analysis
5. How do you ensure the comparisons are fair and how the parameters set? Also, how do you ensure the results are enough to verify the proposal?
6.There is no discussion on the cost effectiveness of the numerical methods surveyed. What is the computational complexity? What is the runtime? Please include such discussions
7. Clearly mention the benefits of chosing tent map by comparing with other chaotic maps

Validity of the findings

no comments

---

## Round 0.2 · Minor Revisions

Dear authors,

Thank you for submitting your article. Based on comments of Reviewer 4 , your article has not yet been recommended for publication in its current form. However, we encourage you to address the minor concerns and criticisms of this reviewer and to resubmit your article once you have updated it accordingly.

Best wishes,

Reviewer 4 ·

Basic reporting

1. In the introduction part, the contributions of this article should be rewritten more clearly. The authors could summarize and list some contribution points.
2.It would be better to incoporate Nomenclature in the beginning of this paper.

Experimental design

There is no discussion on the cost effectiveness of the numerical methods surveyed. What is the computational complexity? What is the runtime? Please include such discussions

Validity of the findings

Figure quality can be improved for Fig 3,4 and 8

Additional comments

nocomments

---

## Round 0.3 · accepted · Accept

Dear authors,

Thank you for clearly addressing all the reviewers' comments and criticisms. All the necessary additions and modifications seem to have been performed. I confirm that the quality of your paper has been improved. The paper now appears to be ready for publication in light of this revision.

Best wishes,

Reviewer 4 ·

Basic reporting

no comments

Experimental design

no comments

Validity of the findings

no comments